# Adaptive multi-frame sampling
# for consistent zero-shot text-to-video editing

**Thérèse Tisseau des Escotais**                    *therese.tisseau-des-escotais@lecnam.net*
*Conservatoire National des Arts et Métiers*
*Ampere Software Technology*

**Clément Rambour**                                  *clement.rambour@isir.upmc.fr*
*Institut des Systèmes Intelligents et Robotique*

**Bertrand Leroy**                                   *bertrand.leroy@ampere.cars*
*Ampere Software Technology*

**Arnaud Breloy**                                    *arnaud.breloy@cnam.fr*
*Conservatoire National des Arts et Métiers*

**Reviewed on OpenReview:** *https://openreview.net/forum?id=vcZ6qdbADL*

## Abstract

Achieving convincing temporal coherence is a fundamental challenge in zero-shot text-to-video editing. To address this issue, this paper introduces AMAC (Adaptive Multi-frame sAmpling for Consistent zero-shot text-to-video editing), a novel method that effectively balances temporal consistency with detail preservation. Our approach proposes a theoretical framework with a fully adaptive sampling strategy that selects frames for joint processing using a pre-trained text-to-image diffusion model. By reformulating the sampling strategy as a stochastic permutation over frame indexes and constructing its distribution based on inter-frame similarities, we promote consistent processing of related content. This method demonstrates superior robustness against temporal variations and shot transitions, making it particularly well-suited for editing long dynamic video sequences, as validated through experiments on DAVIS Perazzi et al. (2016) and BDD100K Yu et al. (2020) datasets. Our code and some examples of generated videos are available in `https://github.com/amac-video-editing/AMAC/`.

## 1 Introduction

State-of-the-art image generation models primarily leverage diffusion processes Ho et al. (2020); Song & Ermon (2019); Song et al. (2020b), which offer enhanced stability and superior detail precision compared to previous generative approaches Goodfellow et al. (2020); Kingma & Welling (2014). Text-to-image (T2I) Rombach et al. (2022) models have achieved remarkable success in generating high-quality visuals by exploiting large-scale multimodal datasets and diffusion-based architectures. These models learn rich representations that effectively align textual descriptions with visual content, enabling fine-grained control Zhang et al. (2023) in image generation. Building upon this advances, diffusion models have been extended to text-driven video synthesis and editing Chai et al. (2023); Wu et al. (2023); Kara et al. (2024). By incorporating temporal modeling, these approaches aim to generate coherent motion while preserving spatial consistency, opening new possibilities for creative content generation and real-world applications in areas such as animation or virtual environments.

Training a text-to-video (T2V) editing model from scratch requires extensive and diverse video datasets with detailed captions, as well as substantial computational resources. Among recent T2V editing methods Guo et al. (2024); HaCohen et al. (2024); Wan et al. (2025); Wang et al. (2025b); Gao et al. (2025); Yang

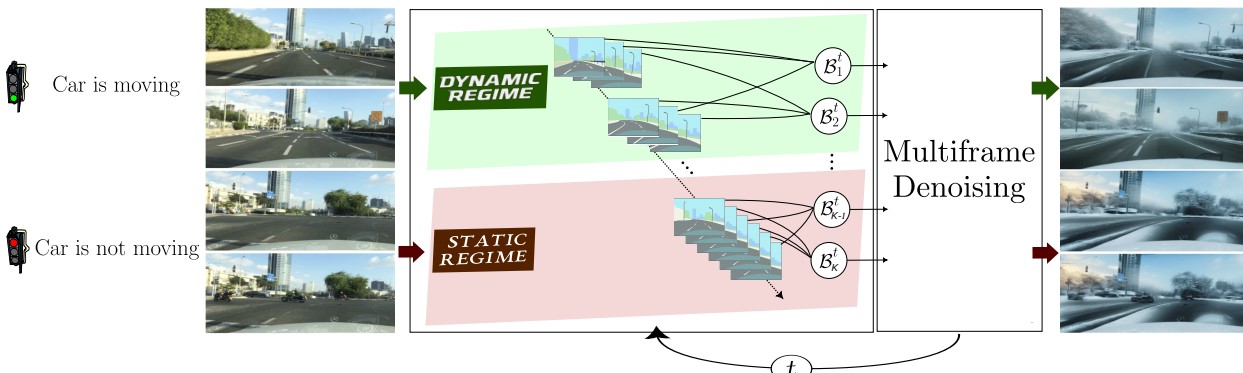

Figure 1: Editing with the prompt "Car driving on a snowy road". AMAC performs multi-frame denoising on batches $\mathcal{B}_k^t$ sampled at each diffusion step. AMAC frame sampling strategy adapts to the dynamics of the video: short-term (resp. long-term) dependencies are leveraged in the dynamic (resp. static) regime.

et al. (2025); Wang et al. (2025a); Zhu et al. (2025); Zhang et al. (2025), only a few provide open-source implementations, such as Guo et al. (2024); HaCohen et al. (2024); Wan et al. (2025). However, Guo et al. (2024) is constrained to 16-frame processing, making it unsuitable for long-duration video editing. HaCohen et al. (2024) and Wan et al. (2025) are also limited to a maximum input length of approximately 100-frame videos. A possibility to extend the length would be to edit videos by block, but without the assurance of long-term consistency. Furthermore, available editing models such as HaCohen et al. (2024); Wan et al. (2025) rely on ~13-14B parameters, making them computationally intensive. As more lightweight alternatives, most current open source video editing methods employ few-shot Chai et al. (2023); Wu et al. (2023); Guo et al. (2024) or zero-shot Kara et al. (2024); Li et al. (2024); Wang et al. (2024) adaptations of T2I models Rombach et al. (2022); Zhang et al. (2023). While few-shot methods lead to interesting results, they require dedicated retraining for each video, making them impractical for editing large sets of long videos.

Conversely, zero-shot methods based on T2I models are less computationally demanding and thus favored for editing long video segments. However, achieving satisfactory zero-shot editing with high temporal consistency between frames remains challenging. Recent works addressing this challenge can be divided into two broad categories: temporal information injection and similarity-based regularization. The first category Kara et al. (2024); Li et al. (2024) processes multiple frames simultaneously to enforce temporal coherence, while the second Li et al. (2024); Wang et al. (2024) leverages natural redundancies within videos by merging similar tokens within sliding windows. Although both approaches promote inter-frame consistency, they often either blend unrelated temporal content or oversimplify frames by removing details and textures.

To ensure a balanced trade-off between temporal consistency and details preservation, we propose AMAC, which leverages adaptive frame sampling in zero-shot video editing. We first establish a framework that formulates zero-shot editing as an approximation of an ideal diffusion model operating on all frames jointly. This approximation processes frame groups sampled at each diffusion timestep. From this perspective, most state-of-the-art methods Kara et al. (2024); Li et al. (2024); Wang et al. (2024) can be viewed as employing fixed or deterministic sampling strategies. We transcend these limitations by proposing an adaptive approach that samples batches based on their similarity. The resulting method thus leverages both short- and long-term temporal dependencies across all frames, achieving the desired balance between temporal smoothness and detail preservation, as illustrated in Figure 1. In short, our main contributions are the following:

- We propose a controlled stochastic strategy for a zero-shot video editing that promotes temporal coherence in a fully adaptive and efficient manner.

- We extensively evaluate our method on two video datasets Perazzi et al. (2016); Yu et al. (2020). In particular, to the best of our knowledge, we are the first to address zero-shot editing on long dynamic video sequences from autonomous driving dataset Yu et al. (2020).

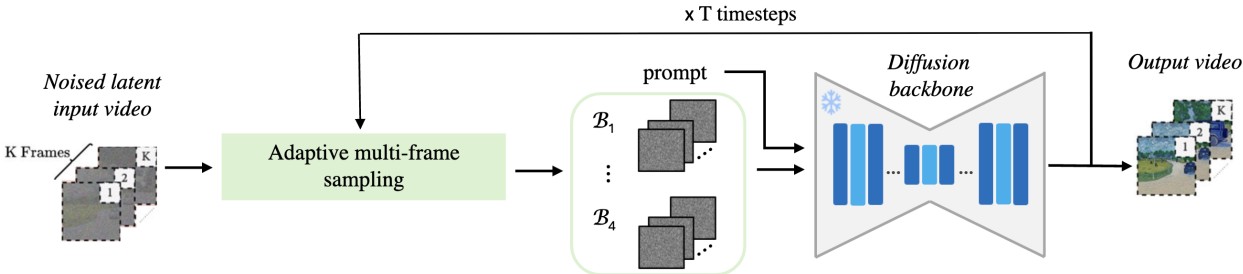

Figure 2: **AMAC denoising process overview.** At each iteration of the diffusion denoising process, frames are permuted using an adaptive stochastic sampling strategy (7, 3) depending on frame similarities (8). The permuted frames are then grouped and jointly denoised by a pre-trained diffusion model.

## 2 Related work

**Text-to-Image Editing**   Text-to-image (T2I) diffusion models have enhanced image editing by enabling fine-grained and controllable modifications through prompt-based guidance Ramesh et al. (2022); Rombach et al. (2022); Saharia et al. (2022); Brooks et al. (2023); Tumanyan et al. (2023), attention manipulation Tumanyan et al. (2023), and feature refinement Saharia et al. (2022); Tumanyan et al. (2023). Some approaches fine-tune pre-trained models on limited examples, enabling customized edits while preserving structural integrity. Others employ training-free strategies, such as manipulating cross-attention maps or optimizing intermediate diffusion features, to apply localized changes without additional model training. Beyond text prompts, control signals like depth or edge maps further enhance editability. However, simply applying these methods to video frame by frame often disrupts temporal coherence.

**Text-to-Video Editing**   Extending T2I diffusion models to video editing presents significant challenges, primarily in maintaining temporal consistency across frames. Despite the frequent release of new video generation methods, video editing progress lags due to the inherent challenge of maintaining input video fidelity. Furthermore, most T2V editing methods are currently not open source, for example Gao et al. (2025); Yang et al. (2025); Wang et al. (2025a); Zhu et al. (2025); Zhang et al. (2025). For the few available methods Guo et al. (2024); Wang et al. (2025b), they are constrained to 16-frame sequences due to their substantial computational cost. For these reasons, common approach involves adapting pre-trained T2I diffusion models by introducing temporal modules Yang et al. (2023); Wu et al. (2023); Ceylan et al. (2023); Qi et al. (2023), enabling frame-by-frame editing while attempting to preserve motion coherence. Some methods integrate spatio-temporal attention layers, while others rely on feature propagation for frame alignment. Alternative approaches enforce consistency by leveraging optical flow models Yang et al. (2023); Cong et al. (2023); Hu & Xu (2023), aligning motion trajectories to minimize flickering. However, these techniques either require additional fine-tuning, depend on external motion estimation, or introduce substantial computational overhead.

**Zero-shot Text-to-Video Editing**   Zero-shot open-source text-to-video (T2V) editing leverages pre-trained T2I diffusion models to modify videos without additional training, though ensuring temporal consistency remains a major challenge. Some approaches, including Pix2Video (Ceylan et al., 2023), FateZero (Qi et al., 2023), and Text2Video-Zero (Khachatryan et al., 2023), enforce consistency through attention blending, cross-frame guidance, or optical flow constraints. TokenFlow (Geyer et al., 2024) takes a different approach by propagating self-attention tokens across frames to enforce structural consistency, mitigating flickering but sometimes producing overly smoothed results. VidToMe (Li et al., 2024) and COVE (Wang et al., 2024) enhance temporal coherence by merging self-attention tokens across frames, reducing flickering and memory overhead. RAVE (Kara et al., 2024) addresses temporal coherence challenge through noise shuffling strategies that preserve motion and semantic structure while enabling diverse edits. AnimateDiff (Guo et al., 2024) enables animation of T2I diffusion models by training a transferable motion module that learns motion priors from video data, with LoRA-based domain adapter. All these methods face challenges in capturing both short- and long-term dependencies from the source video. Token-merging (Bolya et al.,

2023) techniques, while effective at maintaining local stability, struggle to preserve persistent elements, often oversimplifying backgrounds and textures. Conversely, frame shuffling enhances the rendering of static elements and backgrounds by enforcing smoother transitions but tends to introduce flickering during transitions. Our approach addresses these limitations by promoting similar frames grouping through a stochastic process, ensuring both short- and long-term consistency in an efficient manner.

## 3    Preliminaries

**Diffusion models**   Diffusion models Ho et al. (2020); Song et al. (2020a;b) are built upon a forward diffusion process that gradually corrupts an input image into noise and a reverse process that learns to recover the original data distribution through step-by-step denoising.

Given an image $\boldsymbol{x}_0 \sim q(\boldsymbol{x})$ from a real data distribution, the forward process successively adds Gaussian noise over $T$ timesteps according to a variance schedule $\{\beta_t\}_{t=1}^{T}$:

$$q(\boldsymbol{x}_t \mid \boldsymbol{x}_{t-1}) = \mathcal{N}(\boldsymbol{x}_t; \sqrt{1-\beta_t}\boldsymbol{x}_{t-1}, \beta_t\mathbf{I}). \tag{1}$$

To recover data distribution, models learn to retrieve the noise $\epsilon_\theta(\boldsymbol{x}_t, t)$ and are trained to minimize:

$$\mathbb{E}_{\boldsymbol{x}_0, \epsilon, t}\left[\|\epsilon - \epsilon_\theta(\boldsymbol{x}_t, t)\|^2\right], \tag{2}$$

where $\epsilon \sim \mathcal{N}(0, \mathbf{I})$ represents the added noise.

For sample generation given a current noisy estimate $\boldsymbol{x}_t$, the model predicts the noise component $\epsilon_\theta(\boldsymbol{x}_t, t)$, which is then used to approximate $\boldsymbol{x}_{t-1}$. In practice, many applications use a simplified deterministic sampling approach such as DDIM Song et al. (2020a), which allows for faster control over the generation process. The deterministic update equation is given by:

$$\boldsymbol{x}_{t-1} = \sqrt{\bar{\alpha}_{t-1}}\left(\frac{\boldsymbol{x}_t - \sqrt{1-\bar{\alpha}_t}\epsilon_\theta(\boldsymbol{x}_t, t)}{\sqrt{\bar{\alpha}_t}}\right), \tag{3}$$

where $\bar{\alpha}_t = \prod_{s=1}^{t}(1-\beta_s)$ is the cumulative noise schedule product.

**Latent Diffusion Model (LDM)**   Latent Diffusion Models (LDMs) Ramesh et al. (2022); Rombach et al. (2022); Saharia et al. (2022); Sadat et al. shift the diffusion process to the latent space of a pre-trained model. The backbone is typically a Variational Auto-Encoder (VAE) Kingma & Welling (2014) trained on a large dataset for good regularization properties. The encoder $\mathcal{E}$ maps input data $\boldsymbol{x}$ to a lower-dimensional latent representation, expressed as $\boldsymbol{z} = \mathcal{E}(\boldsymbol{x})$. The decoder $\mathcal{D}$ then reconstructs the input from the latent code, ensuring that the output remains perceptually similar to the input $\boldsymbol{x} \simeq \mathcal{D}(\mathcal{E}(\boldsymbol{x}))$.

## 4    Method

This section presents the proposed method based on adaptive frame subsets sampling for T2V editing. The overview of our denoising process is illustrated in Figure 2. The procedure unfolds as follows: at each timestep of the diffusion denoising procedure, we draw a permutation over frame indexes based on their relative similarity. Permuted frames are then grouped into grids, and passed through the pre-trained T2I model. For self-attention steps, redundant tokens are merged, enabling both computational acceleration and temporal consistency enforcement.

### 4.1    Stochastic zero-shot video editing

Let the edited video be designed as an ordered set of $K$ frames with latent representations $\{\boldsymbol{z}^k\}_{k \in \mathcal{I}}$, where $\mathcal{I} = \{1, \cdots, K\}$ denotes the set of frame indexes. We also denote by $\{\mathring{\boldsymbol{z}}^k\}_{k \in \mathcal{I}}$ the latent representations of the original frames. In order to ensure the temporal consistency of the edited video, each frame $\boldsymbol{z}^k$ should

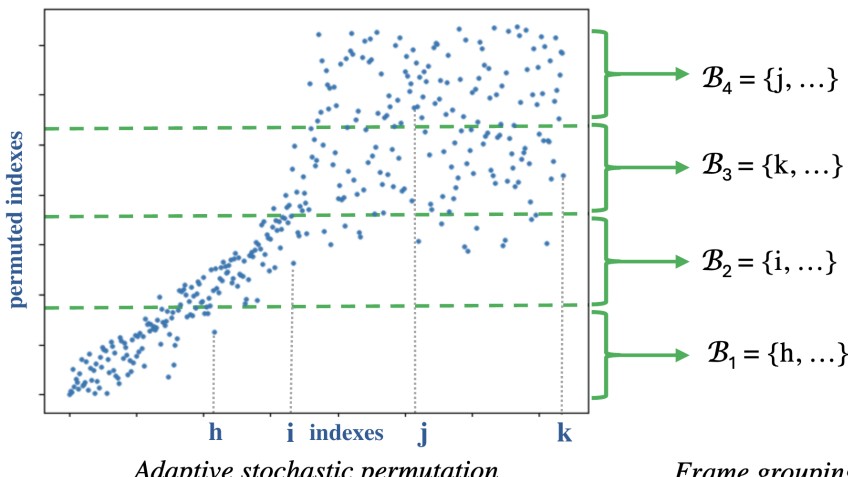

Figure 3: **Adaptive multi-frame sampling.** Sampling batches through a random permutation conditioned by frame similarities (Sec. 4.2). Here, we illustrate the adaptive stochastic permutation for the video shown in Figure 1.

ideally be edited by maximizing its likelihood knowing the prompt $\boldsymbol{\tau}$, all the original representations, and also the other frames being edited, leading to the likelihood:

$$p(\boldsymbol{z}^k \mid \{\mathring{\boldsymbol{z}}^j\}_{j \in \mathcal{I}}, \{\boldsymbol{z}^j\}_{j \in \mathcal{I} \setminus \{k\}}, \boldsymbol{\tau}). \tag{4}$$

From a diffusion perspective, this translates as following a trajectory for each frame from the noisy latent representations $\{\boldsymbol{z}_t^k\}_{1 \leq k \leq K}$ to their clean representations $\{\boldsymbol{z}_0^k\}_{1 \leq k \leq K}$ by injecting the noise model $\epsilon_\theta(\boldsymbol{z}_t^k, \boldsymbol{\tau}, \{\boldsymbol{z}_t^j\}_{j \in \mathcal{I} \setminus \{k\}}, t)$ into equation 1. In the following notations, we drop the conditioning over the original video as it is directly encoded in the starting point $i.e.$ $\forall k \in \mathcal{I}, \boldsymbol{z}_T^k = \mathring{\boldsymbol{z}}_T^k$. Maximizing the likelihood equation 4 by a diffusion process would thus require a model $\epsilon_\theta$ that operates over the concatenation of all tokens constituting all frames of the video. Our goal is to approximate this ideal denoiser for long sequences by leveraging pretrained T2I models.

A reasonable approximation consists in assuming that large proportions of the original video are independent. Allowing to decompose the likelihood equation 4. From the diffusion perspective, this translates into

$$\epsilon_\theta(\boldsymbol{z}_t^k, \boldsymbol{\tau}, \{\boldsymbol{z}_t^j\}_{j \in \mathcal{I} \setminus \{k\}}, t) \simeq \epsilon_\theta(\boldsymbol{z}_t^k, \boldsymbol{\tau}, \{\boldsymbol{z}_t^j\}_{j \in \Omega_k}, t) \tag{5}$$

where $\Omega_k$ is a subset of indexes that represents all the frames that hold relevant temporal inter-dependencies with respect to the frame $k$ (similar objects, continuity of movement, etc.). This theoretically reduces the complexity of the required model, but still does not solve most of the actual practical issues: $i$) The set $\Omega_k$ depends on the original video and the considered frame. It is of course unknown and cannot be trivially obtained (for example, it might consist in non-contiguous elements); $ii$) The sets $\{\Omega_k\}_{k \in \mathcal{I}}$ can have different cardinality, which may be large; $iii$) In a zero-shot editing context, we are restricted to a given backbone with a limited temporal context. Thus, some adaptations are required to mimic the conditioning as stated in equation 5. The points $i$) and $ii$) are addressed in the next paragraph and section 4.2, while point $iii$) is discussed in section 4.3.

In this work, we propose to rely on a stochastic approximation of the latent denoising scheme equation 5. At each timestep $t$ of the diffusion process, we restrict the noise estimation $\epsilon_\theta$ to be computed on random batches $\mathcal{B}_k^t$ of fixed cardinality (typically, a $n \times n$ grid as further described in Section 4.3, with $n = 3$ in our experiments). Thus, we use the denoising scheme

$$\forall k \in \mathcal{B}_k^t, \ \boldsymbol{z}_{t-1}^k \propto \boldsymbol{z}_t^k - \sqrt{1 - \bar{\alpha}_t}\, \epsilon_\theta\big(\{\boldsymbol{z}_t^j\}_{j \in \mathcal{B}_k^t}, \boldsymbol{\tau}, t\big), \tag{6}$$

---

**Algorithm 1** AMAC permutation sampling

---

$\pi(\mathcal{I}) \leftarrow$ ordered set $[0]$
**for** $k = 1$ to $K$ **do**
    Draw random position $i$ according to equation 7 and equation 8
    Insert $k$ at index $i$ in $\pi(\mathcal{I})$
**end for**
**return** Permuted indexes $\pi(\mathcal{I})$

---

where we drop the multiplicative factor $\frac{\sqrt{\bar{\alpha}_{t-1}}}{\sqrt{\bar{\alpha}_t}}$ and in which $\mathcal{B}_k^t$ is sampled each time step using a carefully chosen distribution (see section 4.2). By leveraging this sampling at each step of the diffusion, the embeddings $\{z_0^k\}_{1 \leq k \leq K}$ should then still account for long-range dependencies between all frames.

Interestingly, this proposed formalism allows for re-framing the underlying philosophy of existing zero-shot methods, though it is acknowledged that their implementation can vastly differ from one method to another. Methods relying on sliding-windows Li et al. (2024); Wang et al. (2024) follow the assumption that $\Omega$ can be restricted to broad adjacent frames, leading to the choice $[\![k - b, k + b]\!] \setminus \{k\}$ for $\Omega_k$. Due to computational limitations $b$ is generally too small to capture long term-dependencies. Naive uniform sampling Kara et al. (2024) is efficient for averaging the overall style of the video, but is prone to erasing details or to generating flickers. The issue becomes more prominent for long videos, as the probability to sample related frames vanishes when $K$ increases (*c.f.* supplementary section A.1). This work introduces thus a framework to capture local-global dependencies by adaptively sampling batches $\mathcal{B}_k$ that perform, on average over $t$, an implicit estimation of the sets $\Omega_k$.

## 4.2 AMAC: Adaptive multi-frame sampling

In this section, we aim to obtain an adaptive strategy for drawing the batches $\mathcal{B}_k^t$ in the editing scheme equation 6. Since we want to denoise all frames exactly once at each diffusion step, all frames should be drawn without replacement when forming batches. This problem can be solved by drawing a permutation $\pi$ on frame indexes $\mathcal{I}$, then forming fixed-cardinality sets by grouping consecutive frames on the permuted indexes $\pi(\mathcal{I})$. We require sampled permutations to likely group similar frames together, while still introducing some controlled variability. This can be done by following an iterative insertion scheme, where elements in $\pi(\mathcal{I})$ are added sequentially. Starting from the singleton containing the first instance in $\mathcal{I}$, the $k-$th element is then inserted at the index $i \in \{0, 1, ..., k\}$ with probability:

$$p(k = i) = \frac{f(k, i)}{\sum_{j=0}^k f(k, j)}, \tag{7}$$

where $f$ is a decreasing function with respect to index distance. It is noted that when choosing $f(k, i) = e^{-(1-q)(k-i)}$, we retrieve the so called Mallows distribution of parameter $q$ Mallows (1957) initially proposed for statistical ranking (*c.f.* supplementary material section A.2).

Since ideal frame distribution is unknown, we assume that the trained encoder of the generative model provides a good proxy for the relationships between frames. We use the frame similarity $\text{sim}(\mathring{z}^i, \mathring{z}^j) = \frac{\langle \mathring{z}^i, \mathring{z}^j \rangle}{||\mathring{z}^i|| \, ||\mathring{z}^j||}$ (where $\langle \cdot, \cdot \rangle$ denoting the standard dot product) to guide sampling. This prioritized frame sampling is intended to ensure that each frame is denoised jointly with relevant ones. To do so, we rewrite the function $f$ to account for frame similarity by using the hinge loss on the similarities:

$$\forall (i, j) \in \mathcal{I}^2, f(i, j; d) = \max\big(0, \text{sim}(\mathring{z}^i, \mathring{z}^j) - d\big). \tag{8}$$

In which $d$ is equal to average pairwise frame similarity, to maintain full adaptivity and avoid grouping dissimilar frames. This sampling function ensures that static videos lead to uniform permutations, while dynamic video permutations span restricted, potentially non-contiguous, temporal neighborhoods. The overall permutation sampling strategy is summarized in algorithm 1: it corresponds to Mallows' sampling strategy in which the parameters are adaptively adjusted based on video dynamics, as reflected by inter-frame similarity patterns.

### 4.3 Multi-frame conditioning

Within the proposed editing scheme equation 6, this section presents zero-shot strategies for obtaining multi-frame editing model $\epsilon_\theta\big(\{z_t^j\}_{j\in\mathcal{B}_k^t}, \boldsymbol{\tau}, t\big)$ from a pre-trained T2I model. We stress that the proposed framework is agnostic to these choices and can therefore integrate any method processing a set of frames jointly.

**Grid trick**  To compute the joint denoising of multiple frames, we leverage an efficient strategy developed in recent zero-shot video editing methods Kara et al. (2024). At the beginning of each denoising time-step, the permuted frames are regrouped into grids of $n \times n$ frames. Each grid is then denoised as a 2D image through the pre-trained T2I diffusion model at once, which makes the spatial attention step having a temporal influence on diffusion generation.

**Token-merging**  To speed up the self-attention step and reinforcing the temporal consistency, AMAC model applies a token merging strategy Li et al. (2024); Wang et al. (2024) on each grid which is an efficient attention method leveraging redundancies between tokens. Token merging Bolya et al. (2023) primarily involves eliminating redundant tokens within a sequence based on a given input before self-attention. The output tokens are then duplicated to preserve the input's number of tokens. For this step, we designate one frame in the grid as the reference and eliminate tokens from other frames that exhibit a similarity exceeding a predefined threshold $\lambda$ with the reference tokens. By fixing a threshold rather than a ratio as in Li et al. (2024), token merging strategy is adapting to the current grid needs. More tokens will be merged for highly redundant grids, thus speeding up the self-attention step, while for more diverse grids, fewer tokens will be merged, preserving the variety of details. This copy-paste operation in the denoiser latent space mechanically results into temporal coherence.

The grid trick and token merging strategies work together to enhance temporal coherence and accelerate the overall diffusion process. Adaptive grid sampling enables the model to capture long-range spatio-temporal attention. It also ensures consistency with respect to the source video by associating frames that share common content, thereby guiding the diffusion process. Meanwhile, token merging bias toward oversimplification is mitigated by adequately grouping the frames ensuring efficient editing and strong temporal consistency.

## 5 Experiments

### 5.1 Experimental setup

**Baselines**  We compare our qualitative and quantitative results with four state-of-the-art models : Token-Flow Geyer et al. (2024), VidToMe Li et al. (2024), RAVE Kara et al. (2024) and AnimateDiff Guo et al. (2024). We restrict ourselves to zero-shot baselines with available open-source code or models, maintaining default parameters for all baselines.

**Implementation details**  We use the same backbone for all methods. In particular, the pre-trained diffusion model corresponds to Huggingface Stable Diffusion 1.5 Rombach et al. (2022) T2I model and ControlNet Zhang et al. (2023) with Depth Map (Ranftl et al., 2020) guidance as image-controlling method. We used a fixed $3 \times 3$ grid size and a token merging similarity threshold fixed at 0.8 for all videos. We applied 50 steps of DDIM inversion and 50 steps of denoising iterations for all methods. For TokenFlow, we faced memory limits for videos longer than 100 frames, preventing BDD100K video processing with this baseline. Since AnimateDiff is limited to processing 16-frame videos, we had to truncate the videos to this length for the sake of comparison.

**Datasets**  To standardize quantitative evaluation, we used the dataset and prompts proposed by Kara et al. (2024). The dataset is composed of 21 videos from DAVIS (Perazzi et al., 2016): 15 videos with 36 frames and 6 videos with 90 frames. For each video, there are 2 shape prompts and 4 style prompts, giving 6 editing prompts and 126 text-video pairs. We also evaluate our method on longer, real-world driving videos from BDD100K Yu et al. (2020). Specifically, we sample 4 videos of approximately 350 frames each and apply 8 different realistic style editing prompts (e.g., adding snow or converting to nighttime scenes).

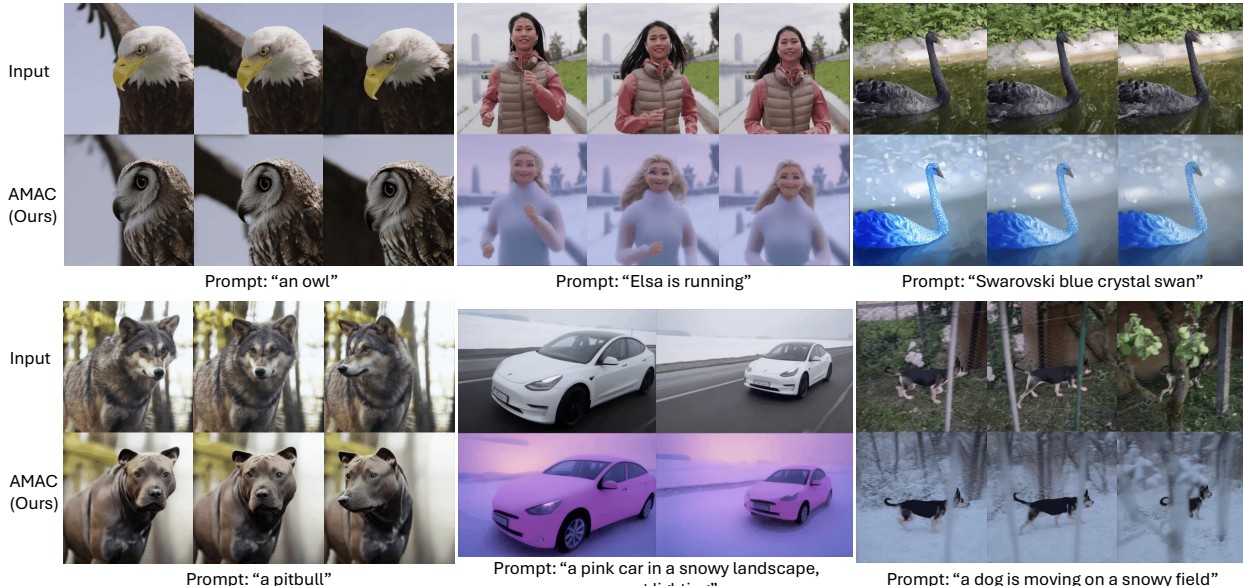

Figure 4: **Qualitative results of AMAC on DAVIS dataset with shape and style prompts.**

Table 1: **Editing scores on DAVIS dataset.**

| Method | Subject Consistency ($\times 10^{-2}$) ↑ | | | CLIP-T ($\times 10^{-2}$) ↑ | | | Warp-SSIM ($\times 10^{-2}$) ↑ | | | $Q_{edit}$ ($\times 10^{-2}$) ↑ | |
|---|---|---|---|---|---|---|---|---|---|---|---|
| | 36-frame | 90-frame | User study | 36-frame | 90-frame | User study | 36-frame | 90-frame | User study | 36-frame | 90-frame |
| TokenFlow | 90.10 | 92.38 | 26,96% | 27.92 | 28.33 | 7,83% | 44.84 | 77.27 | 19,13% | 12.52 | 21.89 |
| VidToMe | 88.10 | 92.48 | **30,00%** | 28.05 | 28.07 | 25,22% | **64.59** | **83.89** | 16,96% | **18.12** | 23.55 |
| RAVE | 91.04 | 93.10 | 16,09% | **29.89** | 30.25 | 32,17% | 50.12 | 76.50 | 30,87% | 14.98 | 23.14 |
| AMAC (Ours) | **91.09** | **94.31** | 26,96% | 29.77 | **30.29** | **34,78%** | 52.49 | 77.93 | **33,04%** | 15.63 | **23.60** |

**Metrics** There is, unfortunately, no standard metric to evaluate video editing. Following prior works Ceylan et al. (2023); Geyer et al. (2024); Yang et al. (2023); Qi et al. (2023); Cong et al. (2023); Li et al. (2024); Kara et al. (2024); Wang et al. (2024), we choose four metrics to evaluate different video editing aspects: Subject Consistency from VBench (Huang et al., 2024) for temporal coherence measurement, CLIP-Text (CLIP-T) for prompt consistency assessment, Warp-SSIM for source video fidelity evaluation, and $Q_{edit}$ for holistic metric combining fidelity and prompt alignment. Subject consistency assesses the persistence of objects with respect to the first and previous frame, so it evaluates temporal coherence but does not take into account the source video and the prompt. CLIP-T computes the similarity between each edited frame and the input prompt, and therefore evaluates the editing but neither the temporal consistency nor the fidelity to input video. Warp-SSIM compares edited video with the warped edited one using the flow of the source video. It measures the fidelity to the input video but tends to neglect temporal consistency and prompt respect. The product $Q_{edit} = WarpSSIM \mathring{u} CLIP-T$ provides a more holistic assessment. We refer the reader to supplementary section A.3 for more details on the metrics. These metrics have inherent limitations Wu et al. (2024): Subject Consistency favors object persistence, CLIP-T is heavily influenced by the CLIP backbone, and Warp-SSIM can be artificially improved by blurred images or smoothed textures. To provide a more complete evaluation, we supplement our analysis with a user study for human assessment. Details on the evaluation protocol are found in supplementary section A.4.

## 5.2 Results

**Short-term editing** Figure 4 presents AMAC editing results with different shape and style prompts on 36-frames and 90-frames DAVIS videos. These results show the ability of our approach to ensure good temporal coherence while conserving details.

Table 2: **Editing scores of AMAC vs. AnimateDiff on 16-frames DAVIS dataset.**

| Method | Subject consistency ↑ ($\times 10^{-2}$) | CLIP-T ($\times 10^{-2}$) ↑ | Warp-SSIM ($\times 10^{-2}$) ↑ |
|---|---|---|---|
| AnimateDiff | 90.33 | 25.75 | 50.74 |
| AMAC (Ours) | **94.51** | **30.05** | **62.48** |

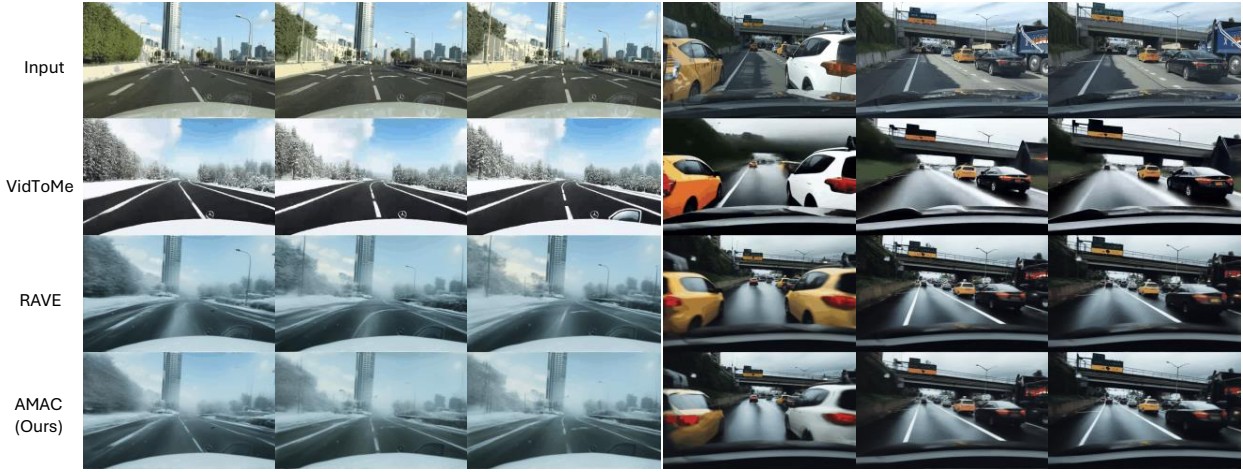

Figure 5: **Qualitative comparison of AMAC and state-of-the-art baselines on BDD100K dataset.** First frames are distant and last frames are adjacent, to illustrate the global coherence (style through time) and the local coherence (details stability).

Table 3: **Editing scores of AMAC vs. baselines on 350-frames BDD100K dataset.**

| Method | Temporal Coherence ($\times 10^{-2}$) ↑ | | Prompt Respect ($\times 10^{-2}$) ↑ | | Source Fidelity ($\times 10^{-2}$) ↑ | |
|---|---|---|---|---|---|---|
| | Subject Consistency | User study | CLIP-T | User study | Warp-SSIM | User study |
| VidToMe | 85.42 | 32,74% | **27.18** | 13,10% | **79.80** | 15,48% |
| RAVE | **90.29** | 16,07% | 25.89 | 25,60% | 74.60 | 23,81% |
| AMAC (Ours) | 89.71 | **51,19%** | 26.05 | **61,31%** | 74.90 | **60,71%** |

Quantitative evaluation in this set-up is displayed in Table 1. AMAC surpasses previous approaches in both temporal consistency and textual alignment especially on 90-frames videos where AMAC shows more than 1-point gain against second-best competitor on subject consistency. Regarding warp-SSIM, a metric that assesses optical flow fidelity, VidToMe achieves unsurprisingly higher scores, given its tendency to oversimplify frame images, losing details and textures in the process. Compared to TokenFlow and RAVE, which maintain higher image level of details, AMAC obtains a better Warp-SSIM score.

For additional evaluation, we compared our method against a recent T2V baseline (AnimateDiff Guo et al. (2024)). Since AnimateDiff is limited to processing 16-frame videos, we had to truncate the DAVIS videos to this length for the sake of comparison. Table 2 and supplementary videos at `https://anonymous.4open.science/r/AMAC-A406` demonstrate our method's superior performance compared to this recent non-zero-shot available baseline.

**Dynamic video editing** Figure 5 shows videos edited by AMAC and baseline methods applied to BDD100K videos of approximately 350 frames using realistic input prompts.

As previously observed, VidToMe, which relies on a high local merging strategy, produces videos with over-simplified textures and reduced detail compared to the input. This oversimplification results in inconsistent

edits, where urban areas transform into forests and only nearby objects are rendered, failing to maintain the intended scene structure. Additionally, the background appears blurry, further diminishing the overall coherence and fidelity of the edited video. RAVE preserves details but introduces significant flickering and style inconsistencies due to its global merging strategy. This results in road lines appearing and disappearing or unintended blending of unrelated elements (for instance, a white car turning yellow after editing). AMAC achieves a visually good trade-off between style and object temporal coherence, reducing inconsistencies across frames. It effectively preserves input video details, even in long sequences, while ensuring adherence to the input prompt.

Quantitative evaluation on long and dynamic videos is presented in table 3. AMAC ranks first across all criteria in the user study, achieving a decisive lead. It outperforms VidToMe in temporal coherence by nearly 20 points and surpasses RAVE in prompt adherence by over 35 points. Additionally, AMAC achieves the highest source fidelity, exceeding both RAVE and VidToMe by nearly 40 points. AMAC is consistently ranked second best method in metric-based evaluations, with the top-ranked method varying. The user study further highlights the limitations of existing metrics in assessing video editing quality. As shown in Table 3, AMAC delivers the best balance between source fidelity, prompt adherence, and temporal coherence.

**Reconstruction**   To evaluate fidelity to the source video, we assess the methods on a reconstruction task, *i.e.*, without input prompt, to quantify their ability to preserve temporal dependencies, fine-grained details and textures. In this setting, since no prompt is provided, CLIP-T is replaced by CLIP-Similarity, which compares frame-by-frame embeddings between the source and edited videos. AMAC surpasses baselines to reconstruction in both temporal consistency and frame-by-frame source fidelity. VidToMe benefits from Warp-SSIM bias but AMAC still outperform TokenFlow and RAVE by a strong margin. See supplementary section A.8 for detailed quantitative and qualitative results.

### 5.3   Additional experiments

Additional experiments are presented here, with more details provided in Appendix A.

**Robustness to abrupt changes.**   We conduct a toy example to highlight the robustness of AMAC to abrupt changes. To simulate quick shot transition, we concatenate four independent 36-frames DAVIS videos and edit this concatenated video with a global style prompt. Figure 6 compares qualitative results of AMAC and its baselines to measure the robustness of models to high changes in the video. As expected, VidToMe removes all details of the source video while RAVE blurs all colors and induces flickering. AMAC keeps details and edits correctly following the different shots, avoiding dramatic frames mixing. More details on this experiment can be found in the supplementary section A.6.

**Sampling strategy and token merging impact.**   AMAC is based on an efficient adaptive sampling strategy implemented using grid trick and token merging. Table 4 assesses the choice of the sampling strategy. AMAC adaptive sampling is compared with two naive sampling strategies: grouping frames using a temporal sliding window and sampling frames from a uniform distribution. For each approach, we also evaluate the impact of token merging within the self-attention step. Local sampling (i.e., sliding window) with token merging effectively reduces to VidToMe, while uniform sampling without merging closely resembles RAVE. Adaptive sampling performs better than local and global strategies, except for Warp-SSIM where local token merging forces a smoother flow between adjacent frames. Moreover, token merging improves temporal coherence and source fidelity for all sampling strategies, but it tends to degrade prompt consistency for naive sampling strategies. The combination of adaptive sampling and token merging manages to improve prompt adherence, highlighting the complementarity of the approaches.

**Token merging threshold**   We conduct an ablation study on the value of the hyperparameter of token merging, and compared these with baselines. We test all possible merging threshold values (from 0.0 to 1.0) with our adaptive sampling strategy. Our method AMAC with threshold 0.8 is the best compromise among all models. For more details, see supplementary section A.5.

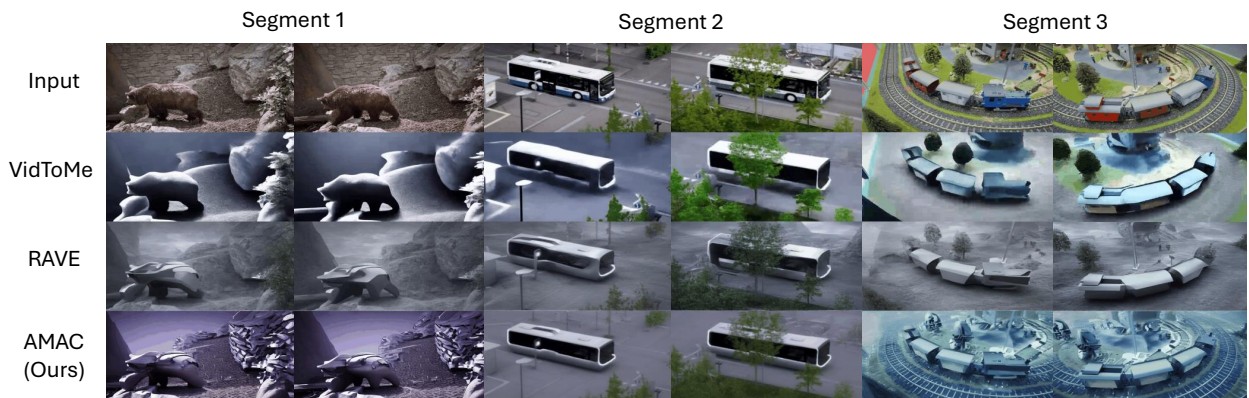



Segment 1     Segment 2     Segment 3

Prompt: "futuristic"



Figure 6: **Qualitative comparison of AMAC and state-of-the-art baselines on a toy example containing abrupt changes.**

Table 4: **Ablation study** on the sampling strategy and token merging on DAVIS 90-frames.

| Sampling | Token Merging | Subject consistency ↑ $(\times 10^{-2})$ | CLIP-T ↑ $(\times 10^{-2})$ | Warp-SSIM ↑ $(\times 10^{-2})$ | $Q_{edit}$ ↑ $(\times 10^{-2})$ |
|---|---|---|---|---|---|
| Local | ✗ | 90.90 | 30.22 | 76.06 | 22.98 |
|  | ✓ | 92.48 | 28.07 | **83.89** | 23.55 |
| Global | ✗ | 93.10 | 30.25 | 76.50 | 23.14 |
|  | ✓ | 93.04 | 30.20 | 77.38 | 23.37 |
| Adaptive | ✗ | 93.87 | 30.20 | 76.17 | 23.00 |
| (Ours) | ✓ | **94.31** | **30.29** | 77.93 | **23.60** |

## 6 Conclusion

In this paper, we propose a zero-shot video editing method based on an adaptive frame sampling strategy during the diffusion process. Our method can easily be plugged into other T2I backbones to enable realistic and temporally coherent video generation, that respects both the input video and the target prompt. Qualitative, quantitative and ablation results demonstrate its applicability to long and highly dynamic videos, showcasing this work's interest. Our approach thus alleviated current limitations of T2I-based video editing paradigms, especially in long term sequences. However, it still faces limitations due to the use of a T2I model: some failure cases are further discussed in the supplementary materials (cf. A.7).

**Broader Impact Statement** While our method is primarily designed for industrial and artistic video editing, we recognize that, like any video generation or editing framework, it could potentially be misused to produce misleading or deceptive content, such as deepfakes. We strongly advocate for the parallel development of robust detection and watermarking tools to mitigate such risks, and we encourage the community to use such methods responsibly. Interested readers can refer to Kundu et al. (2025); Zheng et al. (2025); Wen et al. (2025).

**Acknowledgments** This work was granted access to the HPC resources of IDRIS under the allocation [AD011015263] made by GENCI.

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

## A  Appendix

### A.1  Uniform frame sampling

Let $L$ denote the cardinality of a set $\Omega$ of frames that should be processed jointly because of their shared properties (*e.g.*, all frames containing the same object of short appearance). For one joint diffusion step, we draw a set $\Omega_t$ by sampling $G$ frames from all the $K$ frames of the video. Let $X$ denote the random variable representing the number of frames from $\Omega$ in $\Omega_t$ (*i.e.*, $\text{card}(\Omega \cap \Omega_t)$). Following an uniform sampling strategy for drawing $\Omega_t$ as in Kara et al. (2024), the probability of jointly denoising at least $X \geq 2$ frames from $\Omega$ across $T$ diffusion step is given by:

$$p(X \geq 2) = \left(1 - \frac{K!(K-L)!(K-G)!}{(K-L-G+1)!}\right)^T.\qquad(9)$$

This probability has a quadratic dependence on the length of the video and is inversely proportional to the length $L$ of the considered event. Hence, using uniform frame sampling, the probability of sampling related frames quickly vanishes for long videos.

### A.2  Mallows distribution

The Mallows distribution Mallows (1957) was initially proposed for statistical ranking. Formally, let denote $\mathcal{S}_K$ the set of permutations over $\mathcal{I} = \{1, \cdots, K\}$. For a given permutation $\pi \in \mathcal{S}_K$, an index inversion occurs if we can find two indexes $(i, j) \in \mathcal{I}^2$ such that $i < j$ and $\pi(i) > \pi(j)$. By denoting $\text{inv}(\pi)$ the number of inversions for a permutation $\pi$, Mallows law He et al. (2023) defines a probability for a permutation over an ordered set to be proportional to its number of inversions:

$$p_{\mathcal{M}}(\pi|q) = \frac{q^{\text{inv}(\pi)}}{\sum_{\sigma \in S_n} q^{\text{inv}(\sigma)}}.\qquad(10)$$

The parameter $q$ controls the sampling magnitude of the inversion occurring in $\pi$. In particular, one can prove Mallows (1957) that there exists a constant $c \in \mathbb{R}$ such that:

$$c \cdot \min\{\lambda, K-1\} \leq \mathbb{E}|\pi(s) - s| \leq \min\{2\lambda, K-1\};\qquad(11)$$

with $\lambda = \frac{1}{1-q}$. eq. (11) means that sampling according to Mallows law leads to band permutations where the bandwidth is controlled by the parameter $q$.

As illustrated in fig. 7, when $q \to 0$, the generated permutations are restricted to temporally close frames, and when $q \to 1$, the generated band of permutation matrices widens on average. In the limit cases, we find respectively the identity permutation ($q = 0$) and a uniform random permutation ($q = 1$).

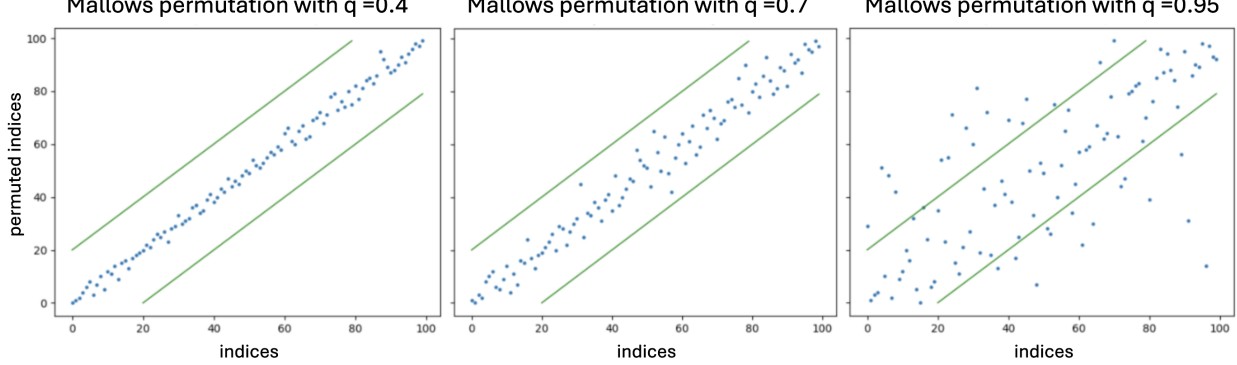

Figure 7: **Examples of Mallows permutation** sampled with different parameters $q$.

## A.3 Metrics

### A.3.1 More details on quantitative metrics

Numerous video editing approaches are evaluated using image editing standard metrics Kara et al. (2024); Wang et al. (2024); Wu et al. (2023), with some of them extended to video evaluation Kara et al. (2024); Couairon et al. (2023), and temporal consistency metrics Wang et al. (2024).

To evaluate temporal consistency, the Subject Consistency metric uses DINO (Caron et al., 2021) features similarity across frames to measure appearance consistency of objects throughout the whole video. Concretely for each frame, the cosine similarity between its features and those of the first frame is computed. This quantity is added to the similarity with its preceding frame. Final metric is obtained by taking the average over all the non-starting video frames Jiang et al. (2024); Chen et al. (2024). Subject Consistency measures the persistence of objects from the first video frame and between adjacent ones. This metric is therefore questionable when applied to long and highly variable videos and may be biased toward global editing strategies that do not account for scene variability.

For textual alignment, the CLIP-T metric calculates the average distance between the CLIP Radford et al. (2021) embedding of each frame and the input prompt. This metric does not take into account the temporal consistency, fidelity to the source video, or the level of details.

To assess the fidelity of the generated video to the source video, Warp-SSIM calculates the average SSIM score Wang et al. (2004) between the warped edited video and the edited video. The warped edited video is the video obtained by applying the flow, calculated by RAFT Teed & Deng (2020) model, of the input video on each edited frame.

For reconstruction evaluation, CLIP-similarity compares corresponding frames of the source and edited videos. It represents the mean of cosine similarity between the embeddings of the corresponding frames.

### A.3.2 Computational cost

Runtime and memory usage for all baselines (TokenFlow, VidToMe and RAVE) are reported alongside our method in Table 5 below for both the DAVIS (36 frames) and BDD100K ($\approx$ 360 frames) datasets. Results show that all methods capable of handling long videos (>100 frames) exhibit comparable computational costs.

Table 5: **Editing scores on DAVIS and BDD100K datasets.**

| Method | **Runtime** (minutes) ↓ | | **Memory** (Mo) ↓ | |
|--------|-------|---------|-------|---------|
|        | DAVIS | BDD100K | DAVIS | BDD100K |
| TokenFlow | 04:36 | - | 134.17 | - |
| VidToMe | **01:11** | **15:44** | 99.67 | 100.46 |
| RAVE | 01:44 | 17:37 | **92.62** | **93.43** |
| Ours | 01:46 | 18:35 | 92.96 | 94.28 |

In addition, we provide a theoretical time complexity study. Let us denote $T$ the number of denoising timesteps, $N$ the number of frames in the video, $g$ the number of frames per grid, $n_t$ the number of tokens per frame, $d_t$ the dimension of a token and $p$ the merging ratio.

Denoising a video frame by frame has a complexity of $\mathcal{O}(T \times N \times (n_t \times d_t)^2)$, but it results in an edited video with no temporal coherence. A naive approach to ensure temporal continuity would be to take all the tokens of a grid centered on the current denoised frame in the self-attention. This naive approach has a complexity of $\mathcal{O}(T \times N \times (g \times n_t \times d_t)^2)$, which is $g^2$ times the complexity of the denoising process frame by frame.

For RAVE, one self-attention operation has a complexity in $\mathcal{O}((g \times n_t \times d_t)^2)$, because the input of the self-attention are all the tokens of a grid as in the naive approach. The self-attention operation is repeated $\frac{K}{g}$ times per timestep, so the total complexity of the denoising process of RAVE is in $\mathcal{O}(T \times N \times g \times (n_t \times d_t)^2)$, which is $\frac{1}{g}$ times the complexity of the naive temporal coherent denoising process.

For VidToMe, there is a small pre-processing step before self-attention operation: calculation of the token which will be merged. This pre-processing operation has a complexity in $\mathcal{O}(((g-1) \times (n_t \times d_t)) \times (n_t \times d_t))$, because there is one frame which is placed in the destination set of merging. Then, the self-attention operation is in $\mathcal{O}(((1 + (g-1) \times (1-p)) \times (n_t \times d_t))^2)$. The self-attention operation is repeated for each of the $N$ frames at each timestep of the reverse process, so the total complexity of the denoising process of VidToMe is in $\mathcal{O}(T \times N \times ((g-1) \times (n_t \times d_t)^2 + (1 + (g-1) \times (1-p)) \times (n_t \times d_t))^2))$, which is the same complexity as the naive temporal coherent denoising process.

For our method AMAC, the computing complexity of one self-attention operation and its pre-processing step with token merging is the same as in VidToMe: $\mathcal{O}((g-1) \times (n_t \times d_t)^2 + (1 + (g-1) \times (1-p)) \times (n_t \times d_t))^2)$. The self-attention operation is repeated one time per grid at each timestep, so the total computing complexity of the denoising process of AMAC is in $\mathcal{O}(T \times \frac{N}{g} \times ((g-1) \times (n_t \times d_t)^2 + (1 + (g-1) \times (1-p)) \times (n_t \times d_t))^2))$, which is about $\mathcal{O}(T \times N \times g \times (n_t \times d_t)^2)$ like RAVE. Thus, our method allows temporal coherence and flickering reduction while having a small time consumption.

Both analyses consistently show that our method offers a favorable trade-off: achieving superior editing quality at roughly equivalent computational cost compared to existing baselines.

## A.4 User study protocol

We conducted two user studies respectively on our BDD100K and DAVIS datasets. In the first one, we presented the participants with three Google Form surveys, each containing four BDD100K videos and prompts, to which 42 anonymous users participated. In the first one, we presented the participants with two Google Form surveys, each containing five DAVIS videos and prompts, to which 46 anonymous users participated. Users are presented with the input prompt, the source video, and three (respectively four) corresponding edited videos (VidToMe, RAVE, AMAC and TokenFlow for short videos), displayed simultaneously in random order. They can watch the videos as many times as they like and adjust the playback speed if desired. They answer three questions, asking them to order the three editing methods with respect to temporal coherence, fidelity to the prompt, and fidelity to the source video. We only display the percentage over the first ranking in the table of results.

## A.5 Ablations

### A.5.1 Ablation on token merging threshold

The goal of this ablation is to measure the impact of the threshold value for the token merging operation. We study it by varying the token merging threshold and comparing AMAC performances with state-of-the-art baselines. To represent VidToMe baseline, we use a low fixed Mallows parameter (0.1) and a low token merging threshold (0.2). Indeed, VidToMe has a local strategy of merging with a high token merging ratio. To represent RAVE baseline, we fix a high Mallows parameter (0.95) and the highest token merging threshold (1.0). Indeed, RAVE has a global shuffling strategy with no token merging.

We see on Figure 8 that VidToMe, which processes a local editing and produces a simplified output, optimizes the prompt fidelity but performs worse at Subject Consistency. We see that RAVE, which keeps source video details and processes a global editing, optimizes Subject Consistency but not prompt respect. Our method AMAC with threshold 0.8 is the best compromise (see prompt respect scale) among all models.

### A.5.2 Ablation on $d$ sampling parameter

To illustrate the sensitivity of the $d$ parameter (Equation 8), we visualize permutation matrix heatmaps computed over 100 samples for several values of $d$ on a challenging scenario: a vehicle that moves dynamically before coming to a stop at a traffic light, producing a clear transition from a dynamic to a static regime around frame 150.

As shown in Figure 9, the choice of $d$ value has a meaningful impact on the resulting permutation. When $d$ is too small (e.g. $Q1$), the permutation approaches a fully random shuffle, disregarding visual similarity entirely. Conversely, when $d$ is too large (e.g. $Q3$ or $Q_{0.9}$), the permutation collapses toward the identity,

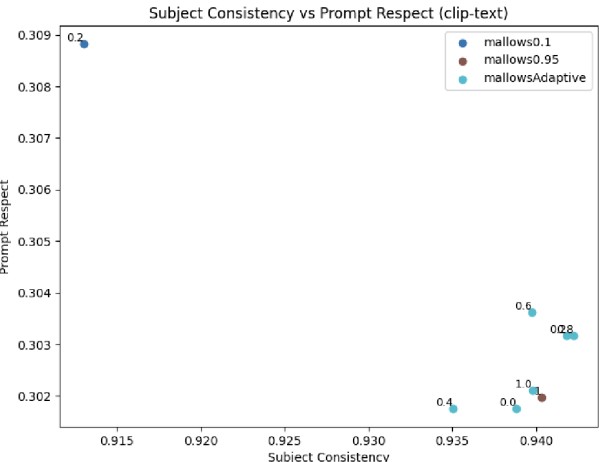

Figure 8: **Ablation plot on token merging threshold.** Scores of prompt fidelity vs temporal coherence for baselines and AMAC with varying token merging threshold.

effectively suppressing any shuffling. The desired behavior (local shuffling during dynamic segments and more global shuffling during static ones) is consistently achieved when $d$ is set close to the average pairwise frame similarity, specifically within the range $[mean - \epsilon, mean + \epsilon]$ (where $\epsilon = 0.01$). This suggests that the method is robust to moderate variations in $d$ around this range, providing practical guidance for practitioners.

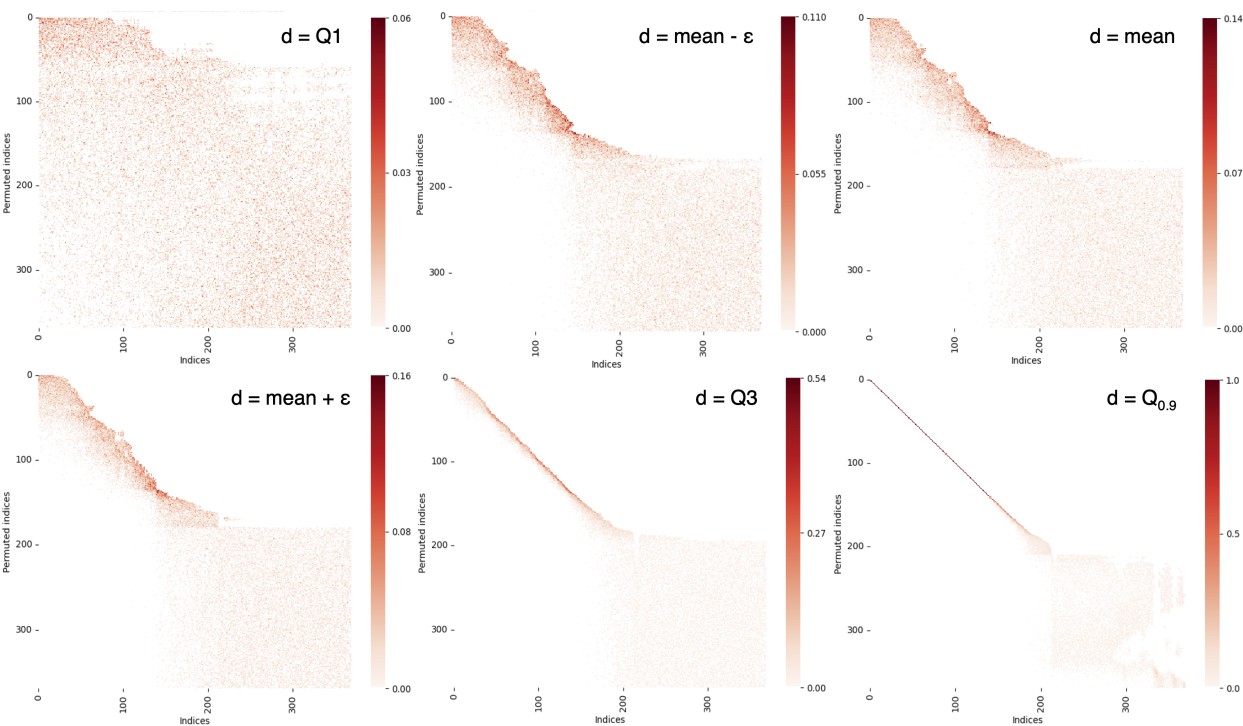

Figure 9: **Heatmap of permutation matrices for ablation on the $d$ parameter.** Sampled permutations adapt to the scenario for $d$ values in a neighborhood of the average pairwise frame similarity (with $\epsilon = 0.01$ here).

### A.5.3 Ablation on DDIM steps

In theory, increasing the number of DDIM steps should improve reconstruction quality by providing a finer approximation of the reverse diffusion process. However, in practice, the performance gains are often marginal while the computational cost increases substantially.

To investigate this, we conducted an ablation study on the number of DDIM denoising steps using all 36-frame DAVIS videos of our dataset. The results reported in Table 6. While Warp-SSIM improves with more steps, Subject Consistency decreases, leading to no clear overall best configuration. At the same time, runtime increases significantly as the number of steps grows. Moreover, using a different number of timesteps than the baselines would result in an unfair comparison.

Increasing the number of DDIM steps introduces two coupled effects: the process becomes closer to DDPM sampling and the number of shuffling operations increases proportionally.

To deeply analyze the results of the ablation, two effects play in augmenting the number of DDIM steps: we are getting closer to DDPM and we are making more shuffling steps. Both factors influence the final performance, making it difficult to isolate which effect drives the observed changes.

Overall, this ablation supports our default choice of 50 DDIM steps as a balanced trade-off between performance and efficiency.

Table 6: **Ablation on DDIM steps on DAVIS with Ours.**

| Denoising steps | Subject consistency ↑ $(\times 10^{-2})$ | CLIP-T $(\times 10^{-2})$ ↑ | Warp-SSIM $(\times 10^{-2})$ ↑ | Runtime (minutes) ↓ |
|---|---|---|---|---|
| 50 | **90.70** | **30.12** | 55.08 | **01:46** |
| 100 | 84.10 | 29.61 | 68.14 | 03:28 |
| 999 | 84.69 | 29.57 | **70.17** | 35:10 |

### A.6 Toy example for robustness evaluation

Our toy example made of four concatenated 36-frames DAVIS videos is a good illustration of how our model AMAC adapts itself to abrupt changes in videos. We can see on Figure 10 that permutations of frames are drawn locally from the same video frame set but globally within each video since the videos are quite static. We can also remark that the last video contains more movement, which implies narrower permutations.

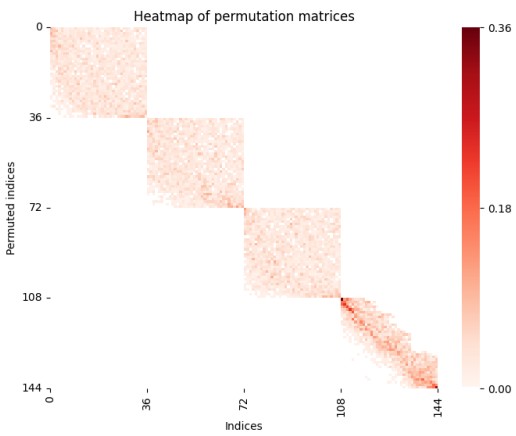

Figure 10: **Heatmap of adaptive permutation matrix on toy example.**

Figure 12 shows more prompt examples on this toy dataset. This confirms that VidToMe simplifies textures and RAVE blurs colors and adds flickering, while AMAC keeps input details and produces temporal coherent outputs. AMAC adapts well to long and dynamic videos.

## A.7    Failure cases

We identify two principal sources of failure: backbone-level reconstruction errors and adaptive shuffling limitations.

During the inversion and denoising process, the T2I backbone may misreconstruct individual frames due to ambiguous latent embeddings. A representative example is a front-facing car reconstructed as a rear-facing car, since both configurations produce similar latent representations, leading to semantic confusion. Our frame grouping with token merging mitigates this by enriching contextual cues, yet cannot fully eliminate such errors. Background blur also originates partly at this level when spatial conditioning is disrupted. Figure 11 (a) illustrates a failure case shared by all methods, including ours: a car whose orientation becomes inconsistent across frames. This reflects the inherent difficulty of preserving directional semantics with a T2I backbone devoid of temporal reasoning.

While our adaptive shuffling strategy promotes global style coherence and improves over uniform stochastic permutation (RAVE), it does not enforce optical flow consistency or directional motion preservation. Failures occur when two globally similar frames contain subtle but semantically opposing details, for instance two cars moving in opposite directions within an otherwise static background. In such cases, the similarity proxy scores both frames as equivalent and groups them together, causing the model to conflate their orientations and produce denoising errors. Figure 11 (b) presents another example in which baseline methods produce an incorrectly oriented car, whereas our method successfully preserves the correct orientation, illustrating that our shuffling strategy seems to resolve some failure edge cases.

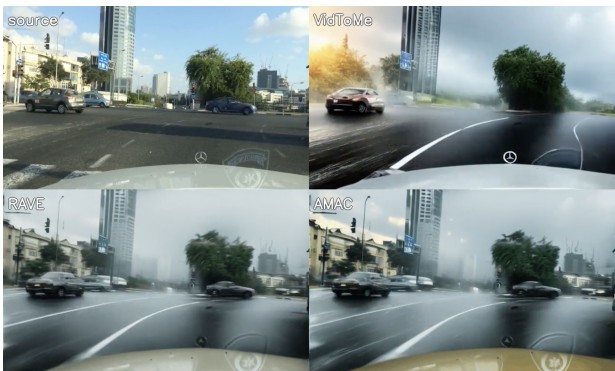 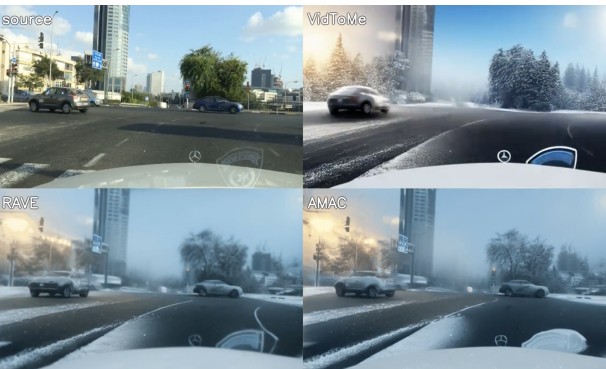

**(a) Shared failure case.** Input prompt: "From a car driving in the rain".

**(b) Corrected failure case.** Input prompt: "Driving on a road with a light, realistic dusting of snow".

Figure 11: Comparison of failure cases.

## A.8    Additional experimental results

**Reconstruction results.**   Figure 13 gives examples of reconstruction videos of baselines and AMAC on 36-frame and 90-frame DAVIS videos. We can observe that TokenFlow gives blurring results and VidToMe simplifies textures while RAVE is globally more faithful to the input. Our method AMAC gives results similar to RAVE with more precise details (jacket texture) and less light flickering. Table 7 shows that AMAC surpasses baselines in the reconstruction task in both temporal consistency and frame-by-frame source fidelity. AMAC is second after VidToMe for flow fidelity, but is far ahead VidToMe for CLIP-similarity. This proves that Warp-SSIM advantages videos which have simple texture and no details.

**More AMAC results on BDD100K dataset.** Figure 15 shows more results of our method AMAC on long and moving BDD100K videos. The top video shows a car turning left at 90° in England. The bottom video shows the road from a car, initially stationary, with another car moving ahead of it, then driving until it reaches a traffic light where it stops again.

AMAC gives detailed results with good temporal coherence and inputs respect.

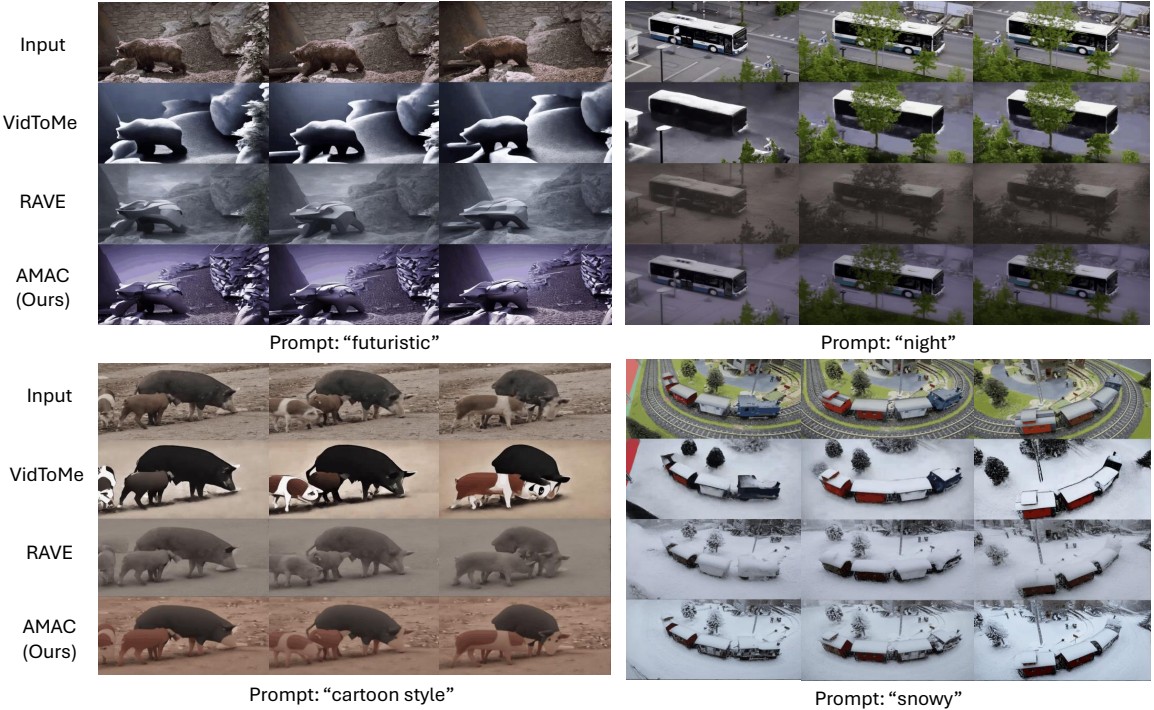

Figure 12: **Qualitative results on toy example with several prompts.**

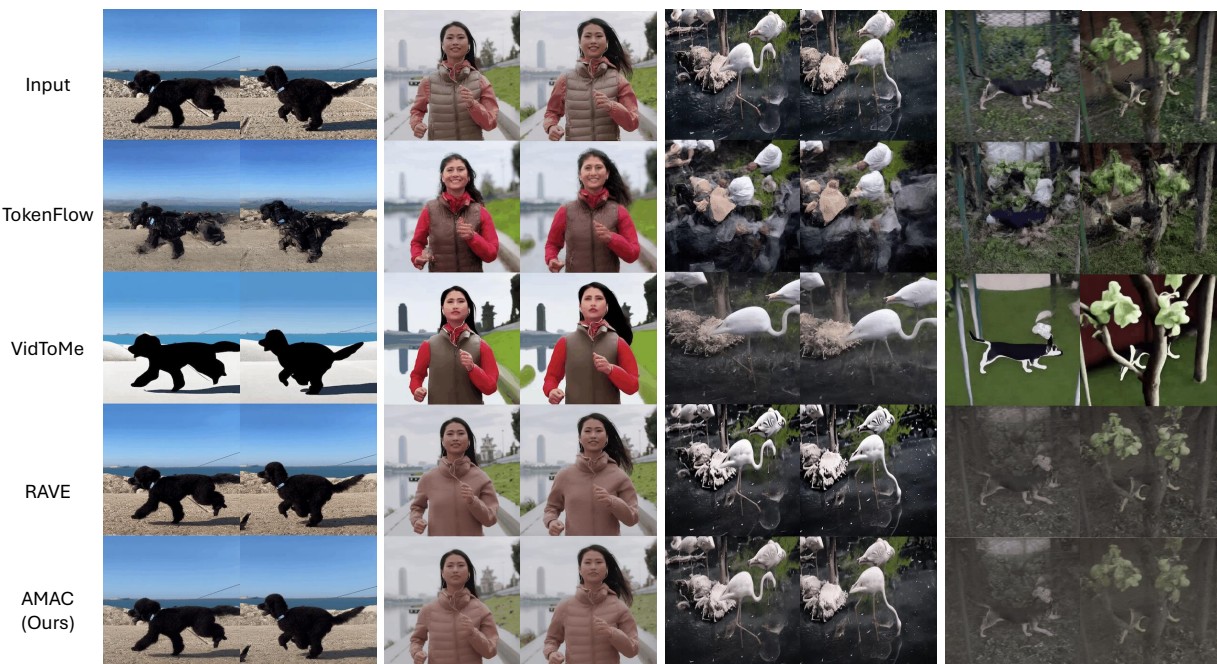

Figure 13: **Qualitative reconstruction results on DAVIS dataset**

Table 7: **Reconstruction scores on DAVIS dataset**

| Method | Subject Consistency ($\times 10^{-2}$) ↑ | | CLIP-similarity ($\times 10^{-2}$) ↑ | | Warp-SSIM ($\times 10^{-2}$) ↑ | |
|---|---|---|---|---|---|---|
| | 36-frame | 90-frame | 36-frame | 90-frame | 36-frame | 90-frame |
| TokenFlow | 88.23 | 93.50 | 81.65 | 81.10 | 45.23 | 75.61 |
| VidToMe | 86.28 | 93.83 | 78.03 | 85.61 | **64.60** | **86.03** |
| RAVE | 89.54 | 95.49 | 85.06 | 85.62 | 49.82 | 78.97 |
| AMAC (Ours) | **89.72** | **95.70** | **85.11** | **86.52** | 55.57 | 80.59 |

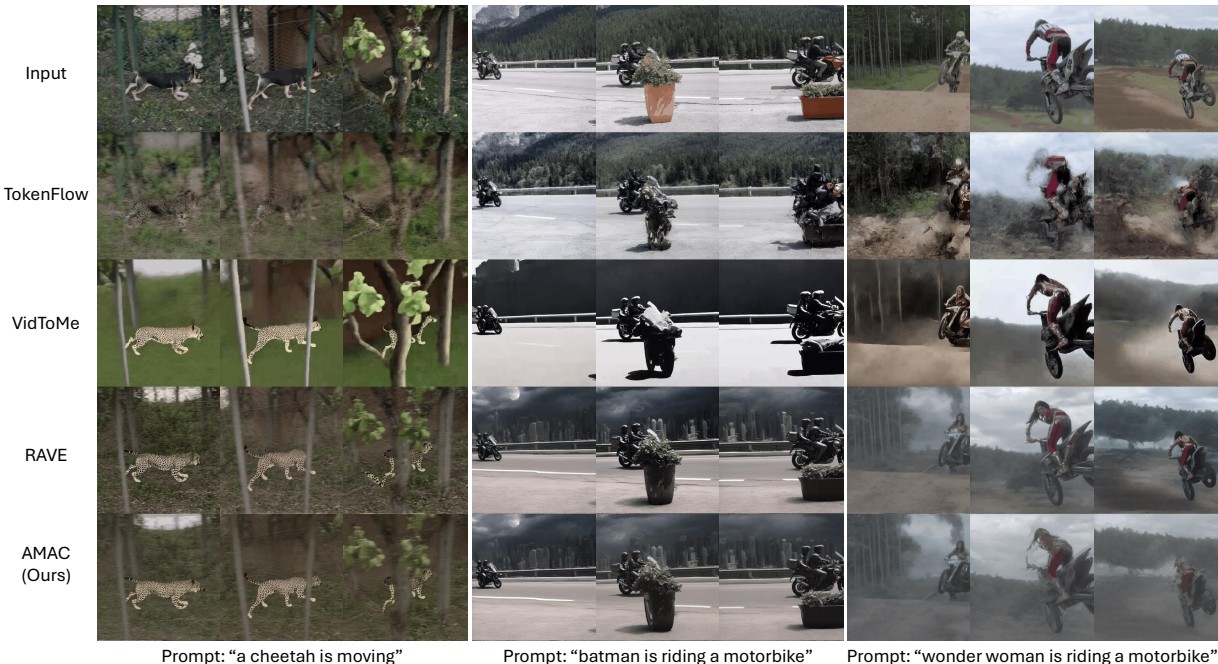

Figure 14: **Qualitative results on DAVIS dataset**

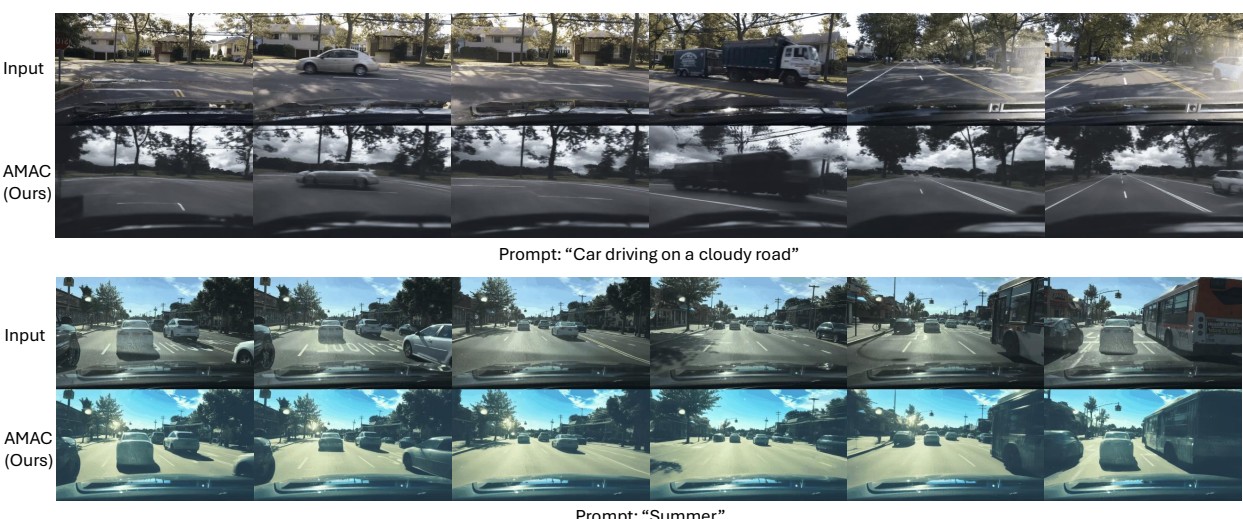

Figure 15: **Qualitative results of AMAC on BDD100K dataset.**

