# OpenReview forum: "Adaptive multi-frame sampling for consistent zero-shot text-to-video editing"
_TMLR — Accepted by TMLR_

### Review · Reviewer_G9KY · 2026-01-30

**Summary Of Contributions:**

This paper proposes AMAC, a zero-shot text-to-video editing framework designed to balance temporal consistency and detail preservation. The core contribution is an adaptive multi-frame sampling strategy that reformulates frame selection as a stochastic permutation based on inter-frame similarities (using a Mallows-like distribution). This sampling allows the model to jointly process related frames (both local and distant) within a pre-trained T2I diffusion model (SD 1.5 + ControlNet) using "grid tricks" and token merging. The authors evaluate the method on DAVIS and BDD100K datasets, claiming superiority in handling long, dynamic sequences.

**Key Strengths**

1. Introduces a theoretically grounded sampling framework (Stochastic Permutation) to address the limitations of fixed-window or uniform sampling.

2. Demonstrates the ability to handle long video sequences (up to 350 frames), which is a challenge for many existing zero-shot methods.

3. Combines adaptive sampling with token merging to improve efficiency and consistency.

**Key Weaknesses:**

1. Semantic and physical inconsistencies: Observation of provided results (e.g., BDD100K samples) reveals significant failures in basic physical logic and motion directionality. For example in the example case:

2. Limited User Study: The human evaluation lacks the scale and diversity needed to support the claim of "decisive lead," especially given the marginal gains in automated metrics.

**Audience:**

Yes

**Audience Explanation:**

The TMLR audience interested in diffusion models and video synthesis would find the attempt to formalize frame sampling as a stochastic process valuable. The discussion on how to approximate global attention in long sequences via sampling is a relevant topic in the current research landscape.

**Broader Impact Concerns:**

No.

**Claims And Evidence:**

No

**Claims Explanation:**

While the paper provides quantitative results showing marginal improvements in Subject Consistency, the qualitative evidence and the nature of the "Adaptive Sampling" raise significant doubts:

1. **Failure in Motion Logic and Source Consistency:** In the BDD100K "car driving in the rain" example, the model edits the car such that its orientation is reversed (the front faces backward while moving forward). This suggests that the adaptive sampling/permutation strategy disrupts the directional semantic priors of the T2I model, leading to "Frankenstein-like" objects that may satisfy frame-wise similarity but violate temporal physics.

2. **Temporal Instability:** The rain/mist in the examples **suddenly disappears**. This indicates that the permutation-based sampling fails to preserve the continuous dynamics of fluid or stochastic elements, likely because these elements do not satisfy the similarity-based grouping logic as effectively as static backgrounds.

3. **Marginal Quantitative Improvement:** In Table 1, the gain in Subject Consistency on 90-frame videos is very small (93.10 vs 94.31). Given the visual artifacts mentioned above, it is questionable whether this statistically represents a meaningful improvement in video quality.

**Requested Changes:**

1. **Address Semantic/Physical Failures**: Provide a detailed analysis or failure case study on why the car orientation and fluid dynamics (rain/mist) fail in long sequences. Explain how the "stochastic permutation" handles (or fails to handle) strong directional motion vectors.

2. **Expand more about User Study**: 14 participants and 10 videos are insufficient for TMLR standards to claim a "decisive lead" in long-video editing which seems inconsistent to the feelling in the example give. Please increase the participant pool and the diversity of video scenes (beyond BDD100K driving scenes).  Or add a "tier" option that the paticipent won't must choose one of them

3. How does the method perform on cases with shot transitions (scene cuts)? This will be a better show cases of long video. Can you demonstrate if the proposed method achieves better consistency in these scenarios compared to baselines?

---

> ### Author Response · Authors · 2026-03-03
> **Answers to reviewer G9KY: general and Requested Change 1**
>
> ### General answer:
>
> We would like to thank Reviewer G9KY for his/her work on this paper.
> Their suggestions and comments (see replies below) have been instrumental in refining the paper.
>
> ---
>
> ### Reviewer G9KY - 1
>
> *1. Address Semantic/Physical Failures: Provide a detailed analysis or failure case study on why the car orientation and fluid dynamics (rain/mist) fail in long sequences. Explain how the "stochastic permutation" handles (or fails to handle) strong directional motion vectors.*
>
> ### Answer G9KY 1
>
> We thank the reviewer for this interesting question.
>
> A key challenge in video editing, especially with text-to-image (T2I) backbone, is maintaining temporal coherence. Since T2I models lack inherent temporal reasoning, they primarily ensure fidelity at the individual frame level rather than enforcing cross-frame semantic or physical consistency. To mitigate this, zero-shot text-to-video (T2V) models use strategies to enforce temporal coherence.
> Our method employs two complementary strategies: frame grid grouping and token merging. However, in certain edge cases, these strategies may prove insufficient to fully preserve temporal coherence. We identify two distinct sources of semantic and physical failures: backbone-level reconstruction errors and stochastic permutation limitations.
>
> During the inversion and denoising process, the T2I backbone may misreconstruct individual frames due to ambiguous embeddings. For instance, a front-facing car may be reconstructed as a rear-facing car because both configurations produce similar latent representations, leading to semantic confusion.
> Our frame grouping strategy with merging of redundant information mitigates this by providing richer contextual cues to the backbone, making such frame-level errors less likely, though not impossible. To addressing this failure edge case, the T2I backbone itself could be improved.
>
> While the stochastic permutation strategy promotes global style coherence across the video, it does not enforce optical flow consistency or directional motion preservation. When frames depicting semantically opposite configurations are grouped together during denoising, semantic drift can occur. Let us consider a scenario where a car is initially driving ahead of the ego-vehicle, and later a different car crosses from right to left. Random frame grouping may place these two frames together, causing the model to conflate their orientations.
> Our adaptive shuffling partially addresses this by grouping visually similar frames, increasing the likelihood of detail persistence. Nevertheless, failures can still occur when two globally similar frames contain subtle but semantically opposing details, for example two cars moving in opposite directions within an otherwise static scene.
>
> These two failure sources explain why car orientations may become inconsistent and fluid dynamics (e.g., rain or mist) may lose physical plausibility in long sequences, where cumulative motion and directional cues play an increasingly important role.
> We will mentioned these limitations in the conclusion of the main paper and dedicate a paragraph in the Section A.7 ("Failure cases") of the Appendix.

---

> ### Author Response · Authors · 2026-03-03
> **Answers to reviewer G9KY: Requested Changes 2 and 3**
>
> ### Reviewer G9KY - 1
>
> *2. Expand more about User Study: 14 participants and 10 videos are insufficient for TMLR standards to claim a "decisive lead" in long-video editing which seems inconsistent to the feeling in the example give. Please increase the participant pool and the diversity of video scenes (beyond BDD100K driving scenes). Or add a "tier" option that the participant won't must choose one of them.*
>
> ### Answer G9KY 2
>
> We clarify that our user study on BDD100K videos involved 42 participants (not 14), evaluating 32 prompt-video pairs, providing a more substantial evaluation than initially perceived.
> Still, to address concerns about scene diversity, we conducted an additional user study on DAVIS videos with 46 participants, extending our evaluation beyond driving scenarios to a broader range of video content.
> Regarding the suggestion of a "tier" option, we acknowledge that such an option could provide additional nuance. However, the forced-choice paradigm is standard practice in perceptual evaluation studies, as it reduces the bias introduced by participants defaulting to neutral responses and yields clearer differentiation between methods.
>
> The results of this additional user study are summarized in Table below. Our method achieves the best performance in terms of source fidelity and prompt respect, and is on par in temporal coherence on this dataset with VidToMe, which scores considerably lower on source fidelity, suggesting a fidelity-coherence trade-off.
> Differences between methods are naturally smaller on the DAVIS dataset, which contains short, simple and object-centric videos, which makes it harder for human evaluators to perceive subtle differences in editing quality and temporal consistency compared to the more complex and dynamic scenarios of BDD100K dataset.
>
> Taken together, both user studies reinforce the conclusion that our method represents a good trade-off among zero-shot video editing approaches.
> We will incorporate these additional results into Table 1 of the main paper and Section A.4 of the Appendix.
>
>
> **Table: User study on DAVIS dataset.**
>
> |              | Source fidelity | Prompt respect | Temporal coherence |
> |--------------|-----------------|----------------|--------------------|
> | TokenFlow    | 19,13%          | 7,83%          | *26,96%*      |
> | VidToMe      | 16,96%          | 25,22%         | **30,00%**         |
> | RAVE         | *30,87%*   | *32,17%*  | 16,09%             |
> | Ours         | **33,04%**      | **34,78%**     | *26,96%*      |
>
>
> ---
>
> ### Reviewer G9KY - 3
> *3. How does the method perform on cases with shot transitions (scene cuts)? This will be a better show cases of long video. Can you demonstrate if the proposed method achieves better consistency in these scenarios compared to baselines?*
>
> ### Answer G9KY 3
> Indeed, shot transitions represent a challenging scenario for zero-shot T2I-based video editing, as abrupt scene changes introduce strong visual discontinuities that are difficult to handle without temporal priors.
> This is precisely why we specifically addressed this case of scene cuts in Section 5.3 of the main paper ("Additional experiments"), specifically in the paragraph titled "Robustness to abrupt changes". Additional implementation details are provided in Section A.6 ("Toy example for robustness evaluation") of the Appendix. Furthermore, we include two long videos demonstrating scene cuts handling in the *anonymous* repository
> https://anonymous.4open.science/r/AMAC-A406 under the directory "Shot transitions".
>
> To further illustrate our method's advantages on long videos with shot transitions, we have added an additional example to the GitHub repository showing a car making a 90° left turn. This example demonstrates that baseline methods suffer from inconsistent road line markings, either removing them entirely or causing significant flickering, while our method maintains more consistent and persistent road markings throughout the turn.
>
> These examples collectively show that our similarity-based permutation strategy helps maintain visual consistency even across abrupt scene changes and sharp directional transitions, which are common shot transition challenges in long-video editing scenarios.
>
> Following this question, we added a video to GitHub repository.

---

### Review · Reviewer_Akyj · 2026-02-08

**Summary Of Contributions:**

This paper proposed a zero-shot video editing method called AMAC that wraps a standard T2I diffusion model and makes it denoise groups of frames together, with a clever adaptive way of choosing those groups.

Strength
S1) Training-free methods always appreciated.

S2) Handles long, dynamic videos with a T2I backbone, while many T2V editors are restricted to ~16 frames or require per‑video tuning.

S3) Overall the paper reads smoothly.

Weakness
W1) The background in edited videos by AMAC are blurry, possibly due to token merging.

**Audience:**

Yes

**Audience Explanation:**

Video editing is a hot topic. Also this method is training free, likely would be beneficial to other works as well.

**Broader Impact Concerns:**

/

**Claims And Evidence:**

Yes

**Claims Explanation:**

Overall the evaluation looks convincing, good choice of usage in DINO for subject consistency. The claims also look valid and did not overclaim.

**Requested Changes:**

C1) Consider adding quantitative runtime / memory analysis on comparisons vs VidToMe/RAVE/TokenFlow, especially on long clips to support the claim of efficiency.

C2) Consider redrawing figure 1 to make it one column to double column. Current layout is a bit unconventional.

C3) Explicit discussion on blurry background issues in AMAC.

C4) "The paper is organized as follows: related work (section 2), preliminaries (section 3), method (section 4),
experiments (section 5), and conclusions (section 6)." This line feels not necessary. Consider removing it for space.

C5) Consider to ablate with 1000 steps DDIM, which should give better reconstruction latent then 50 steps DDIM? Will it improve the results?

C6) In evaluation, consider adding CLIP-I for neighbor frames to evaluate temporal smoothness, which is commonly used as a metric in other video editing works.

---

> ### Author Response · Authors · 2026-03-03
> **Answers to reviewer Akyj: general and Requested Changes C1 to C4**
>
> ### General answer:
> We would like to thank Reviewer Akyj for his/her work on this paper.
> Their suggestions and comments (see replies below) have been instrumental in refining the paper.
>
> ---
>
> ### Reviewer Akyj - C1
> *C1) Consider adding quantitative runtime / memory analysis on comparisons vs VidToMe/RAVE/TokenFlow, especially on long clips to support the claim of efficiency.*
>
> ### Answer Akyj C1
> Runtime and memory usage for all baselines (TokenFlow, VidToMe and RAVE) are reported alongside our method in Table below for both the DAVIS (36 frames) and BDD100K ($\approx 360$ frames) datasets.
> Results show that all methods capable of handling long videos (>100 frames) exhibit comparable computational costs.
>
> In addition, we provide a theoretical time complexity study, which is too long to be put here, but that we will add to the Appendix with the above table in Section A.3.2 ("Computational cost").
> Both analyses consistently show that our method offers a favorable trade-off: achieving superior editing quality at roughly equivalent computational cost compared to existing baselines.
>
> **Table: Editing scores on DAVIS and BDD100K datasets.**
>
> | Method     | Runtime (min) ↓ |        | Memory (Mo) ↓ |        |
> |------------|-----------------|--------|---------------|--------|
> |            | DAVIS          | BDD100K | DAVIS        | BDD100K |
> | TokenFlow  | 04:36          | -      | 134.17       | -      |
> | VidToMe    | **01:11**      | **15:44** | 99.67     | 100.46 |
> | RAVE       | 01:44          | 17:37  | **92.62**    | **93.43** |
> | Ours       | 01:46          | 18:35  | 92.96        | 94.28  |
>
> ---
>
> ### Reviewer Akyj - C2
> *C2) Consider redrawing figure 1 to make it one column to double column. Current layout is a bit unconventional.*
>
> ### Answer Akyj C2
> The redrawn Figure 1 has been incorporated into the main paper.
>
> ---
>
> ### Reviewer Akyj - C3
> *C3) Explicit discussion on blurry background issues in AMAC.*
>
> ### Answer Akyj C3
> The stochastic frame shuffling applied at each timestep of the diffusion denoising process promotes global style coherence across the generated video. However, because it perturbs the model’s spatial conditioning, it may occasionally interfere with precise spatial reasoning, which can manifest as slight background blur in some cases.
> Regarding token merging, its effect is not blur but texture simplification, producing smoother outputs.
>
> Our adaptive shuffling strategy mitigates this artifact compared to RAVE by controlling the degree of shuffling. This improvement is supported quantitatively by the reported evaluation scores.
>
> We will explicitly add this discussion in Section A.7 ("Failure cases") of the Appendix, also asked by the reviewer G9KY.
>
> ---
>
> ### Reviewer Akyj - C4
> *C4) "The paper is organized as follows: related work (section 2), preliminaries (section 3), method (section 4), experiments (section 5), and conclusions (section 6)." This line feels not necessary. Consider removing it for space.*
>
> ### Answer Akyj C4
> The quoted line in the introduction has been removed from the main paper.

---

> ### Author Response · Authors · 2026-03-03
> **Answers to reviewer Akyj: Requested Changes C5 and C6**
>
> ### Reviewer Akyj - C5
> *C5) Consider to ablate with 1000 steps DDIM, which should give better reconstruction latent then 50 steps DDIM? Will it improve the results?*
>
> ### Answer Akyj C5
> We thank the reviewer for this suggestion.
>
> In theory, increasing the number of DDIM steps should improve reconstruction quality by providing a finer approximation of the reverse diffusion process. However, in practice, the performance gains are often marginal while the computational cost increases substantially.
>
> To investigate this, we conducted an ablation study on the number of DDIM denoising steps using all 36-frame DAVIS videos of our dataset. The results reported in Table below.
> While Warp-SSIM improves with more steps, Subject Consistency decreases, leading to no clear overall best configuration. At the same time, runtime increases significantly as the number of steps grows.
> Moreover, using a different number of timesteps than the baselines would result in an unfair comparison.
>
> Increasing the number of DDIM steps introduces two coupled effects: the process becomes closer to DDPM sampling and the number of shuffling operations increases proportionally.
>
> To deeply analyze the results of the ablation, two effects play in augmenting the number of DDIM steps: we are getting closer to DDPM and we are making more shuffling steps. Both factors influence the final performance, making it difficult to isolate which effect drives the observed changes.
>
> Overall, this ablation supports our default choice of 50 DDIM steps as a balanced trade-off between performance and efficiency.
> We will include this study in Section A.5.3 ("Ablation on DDIM steps") of the Appendix.
>
>
> **Table: Ablation on DDIM steps on DAVIS with Ours.**
>
> | Denoising steps | Subject consistency (×10⁻²) ↑ | CLIP-T (×10⁻²) ↑ | Warp-SSIM (×10⁻²) ↑ | Runtime (minutes) ↓ |
> |-----------------|--------------------------------|------------------|----------------------|----------------------|
> | 50              | **90.70**                      | **30.12**        | 55.08               | **01:46**            |
> | 100             | 84.10                          | 29.61            | 68.14               | 03:28                |
> | 999             | 84.69                          | 29.57            | **70.17**           | 35:10                |
>
>
> ---
>
> ### Reviewer Akyj - C6
> *C6) In evaluation, consider adding CLIP-I for neighbor frames to evaluate temporal smoothness, which is commonly used as a metric in other video editing works.*
>
> ### Answer Akyj C6
> CLIP-I measures the average pairwise cosine similarity between CLIP embeddings of adjacent video frames and is widely used as a proxy for temporal smoothness in prior video editing works. Since CLIP is trained with image-text contrastive learning, it primarily captures global semantic information when assessing similarity between two images.
> In contrast, the Subject Consistency metric, also commonly used in the video editing community, measures the average pairwise similarity between DINO embeddings of adjacent frames. As DINO is trained via self-supervised learning without language supervision, it tends to capture finer-grained visual and structural details, enabling a more localized comparison between frames.
>
> To compare these two metrics, we evaluated both on the DAVIS dataset and reported results in the Table below. They both seem to evaluate method quite similarly, with larger differences for Subject Consistency.
>
> Since both metrics aim to assess temporal coherence but rely on different visual representations, and given that DINO-based similarity is more aligned with our image-to-image evaluation protocol, we choose to retain only the Subject Consistency metric in the main paper. This also allows us to maintain one metric per evaluation criterion (source fidelity, prompt alignment, and temporal coherence) for clarity and conciseness.
>
>
> **Table: CLIP-I vs Subject Consistency scores on DAVIS.**
>
> | Method     | CLIP-I (×10⁻²) ↑ | Subject Consistency (×10⁻²) ↑ |
> |------------|------------------|-------------------------------|
> | TokenFlow  | 96.35            | 90.10                         |
> | VidToMe    | 96.93            | 88.10                         |
> | RAVE       | 97.09            | 91.04                         |
> | Ours       | **97.19**        | **91.09**                     |

---

### Review · Reviewer_RNJJ · 2026-02-18

**Summary Of Contributions:**

The paper introduces AMAC (Adaptive Multi-frame sAmpling for Consistent Zero-Shot Text-to-Video Editing), a framework designed to improve temporal coherence in zero-shot text-to-video (T2V) editing.

Key contributions include:

 - Theoretical Framework: Reformulating zero-shot editing as a stochastic approximation of an ideal joint-frame diffusion process.

 - Adaptive Sampling Strategy: A novel method that uses a stochastic permutation (inspired by the Mallows distribution) conditioned on inter-frame similarity to group related frames for joint denoising.

    - Dynamic Adaptation: The strategy automatically adjusts to video dynamics, leveraging short-term dependencies for high-motion "dynamic" regimes and long-term dependencies for "static" regimes.

 - Adaptive Token Merging: Utilizing a fixed similarity threshold to merge redundant tokens, which preserves detail in diverse frames while accelerating computation in redundant ones.

 - Long-Sequence Benchmarking: Extensive evaluation on the BDD100K dataset, specifically addressing long autonomous driving sequences.

Strengths:
  - Zero-Shot (no training required, not even one-shot)

  - Mindful reformulation that takes into consideration the motion within frames to adapt towards providing better conditions to the underlying T2I model.

Weaknesses:
   - Does not discuss the choice between T2I-based video models vs open-source T2V models (such as WAN, LTX, etc.). I am aware of their limitations, but I do think it deserves discussion rather than a single line.

   - Many of the cited works are not available and thus not easily comparable. This makes it harder to read, since the discussion runs alongside the cited works, but the results don't compare to them.

   - Weak Proxy for frame similarity: Based on my understanding, the frame similarity is the key guiding feature in AMAC. Wouldn't it be better if a more semantic feature were used for this choice, such as CLIP similarity?

**Audience:**

Yes

**Audience Explanation:**

The TMLR audience includes researchers focused on generative modeling, diffusion processes, and computer vision. AMAC contributes a novel perspective on stochastic sampling within the sampling theory for conditional generation.

**Broader Impact Concerns:**

The paper does not explicitly include a Broader Impact Statement. While it is a technical contribution to video editing, text-to-video models carry inherent risks regarding deepfakes and the generation of misleading content.

Recommendation: The authors should add a brief statement acknowledging that while AMAC improves creative editing, it could potentially be used to create more convincing synthetic videos, necessitating the continued development of watermarking or detection tools.

**Claims And Evidence:**

No

**Claims Explanation:**

The authors provide the following evidence: quantitative metrics, qualitative comparisons, and a human user study.

Positives:
   - Quantitative evaluation shows certain improvements with AMAC in some utilities. Albeit some older methods also score better in other tasks.
   - Qualitative and Human-study also show some improvements.

Negatives:
   - Ablation only shows quantitative numbers. Why not use a human study to prove this point as well, if quant metrics are not that reliable?
   - Comparison to very recent methods, such as those mentioned in this line: "Furthermore, most T2V editing methods are currently not open source, for example, Gao et al. (2025); Yang et al. (2025); Wang et al. (2025a); Zhu et al. (2025); Zhang et al. (2025)", is very important for understanding the impact of this contribution. This makes it very difficult to make a strong decision.
   - Using SD1.5 in 2025-2026 is very outdated and makes it even harder to understand the scaling of the properties discussed in the paper. Can it be applied to newer models while getting the same benefits? Unsure!
   - Explanation of the choice of similarity proxy (similarity between VAE encoded frames).

**Requested Changes:**

Critical for Acceptance:

  - Algorithmic Complexity: Please include a brief discussion or a table comparing the computational overhead (inference time/VRAM usage) of AMAC's adaptive sampling versus the fixed-window "Local" sampling. This is vital for users prioritizing speed.

  - Mathematical Clarification: In section 4.1, the transition from equation (5) to (6) is clear, but explicitly stating the range of the cardinality of $\mathcal{B}_{k}^{t}$ (used in the experiments) earlier in the text would improve readability.

Strengthening the Work:

   - Failure Cases: It would be insightful to see a "Failure Cases" section in the Appendix. When does the adaptive similarity proxy fail? (e.g., extremely low-light or repetitive textures).

   - Hyperparameter Sensitivity: While an ablation on the token merging threshold is provided, a brief comment on the sensitivity of the $d$ parameter in the sampling function (equation 8) would be helpful for practitioners.

   - Explanation/discussion of the weaknesses mentioned in the previous answers

---

> ### Author Response · Authors · 2026-03-03
> **Answers to reviewer RNJJ: general and Requested Changes RC1 to RC4**
>
> ### General answer:
> We would like to thank Reviewer RNJJ for his/her work on this paper.Their suggestions and comments (see replies below) have been instrumental in refining the paper.
>
> ---
>
> ### Reviewer RNJJ - RC1
> *RC1) Algorithmic Complexity: Please include a brief discussion or a table comparing the computational overhead (inference time/VRAM usage) of AMAC's adaptive sampling versus the fixed-window "Local" sampling. This is vital for users prioritizing speed.*
>
> ### Answer RNJJ RC1
> We thank the reviewer for raising this important practical concern. This point was similarly raised by reviewer Akyj (C1), and we refer the reviewer to our shared response for the full discussion.
> In short, we have added a runtime and memory comparison table alongside a theoretical time complexity analysis in Section A.3.2 ("Computational cost") of the Appendix.
>
> ---
>
> ### Reviewer RNJJ - RC2
> *RC2) Mathematical Clarification: In section 4.1, the transition from equation (5) to (6) is clear, but explicitly stating the range of the cardinality of $\mathcal B_k^t$ (used in the experiments) earlier in the text would improve readability.*
>
> ### Answer RNJJ RC2
> We will explicitly add in Section 4.1 a clarification specifying that the cardinality of $\mathcal B_k^t$ is equal to the grid size, which is set to 9 in our experiments.
>
> ---
>
> ### Reviewer RNJJ - RC3
> *RC3) Failure Cases: It would be insightful to see a "Failure Cases" section in the Appendix. When does the adaptive similarity proxy fail? (e.g., extremely low-light or repetitive textures).*
>
> ### Answer RNJJ RC3
> Similar concerns were raised by reviewers G9KY (1) and Akyj (C3), and we refer the reviewer to our shared response addressing failure cases arising from backbone limitations and the shuffling strategy.
>
> Regarding failures specific to the adaptive similarity proxy: errors can still occur when two globally similar frames contain subtle but semantically opposing details, for example two cars moving in opposite directions within an otherwise static scene. In such cases, both frames are assessed as globally similar by the proxy and grouped together, despite their opposing semantic content, which can lead to semantic confusion and output denoising errors.
>
> You will find more details and discussions about failure cases the Section A.7 ("Failure cases") of the Appendix.
>
> ---
>
> ### Reviewer RNJJ - RC4
> *RC4) Hyperparameter Sensitivity: While an ablation on the token merging threshold is provided, a brief comment on the sensitivity of the $d$ parameter in the sampling function (equation 8) would be helpful for practitioners.*
>
> ### Answer RNJJ RC4
> We thank the reviewer for this insightful suggestion.
>
> To illustrate the sensitivity of the $d$ parameter (Equation 8 of the main paper), we visualize permutation matrix heatmaps computed over 100 samples for several values of $d$ on a challenging scenario:
> a vehicle that moves dynamically before coming to a stop at a traffic light, producing a clear transition from a dynamic to a static regime around frame 150.
>
> As shown in Figure 9 (added in Section A.5.2 of the paper), the choice of $d$ value has a meaningful impact on the resulting permutation.
> When $d$ is too small (e.g. $Q1$), the permutation approaches a fully random shuffle, disregarding visual similarity entirely.
> Conversely, when $d$ is too large (e.g. $Q3$ or $Q_{0.9}$), the permutation collapses toward the identity, effectively suppressing any shuffling.
> The desired behavior (local shuffling during dynamic segments and more global shuffling during static ones) is consistently achieved when $d$ is set close to the average pairwise frame similarity, specifically within the range $[ mean - \epsilon, mean + \epsilon ]$ (where $\epsilon = 0.01$).
> This suggests that the method is robust to moderate variations in $d$ around this range, providing practical guidance for practitioners.
>
> This ablation discussion and its discussion will be incorporated in Section A.5.2 ("Ablation on $d$ sampling parameter") of the Appendix.

---

> ### Author Response · Authors · 2026-03-03
> **Answers to reviewer RNJJ: Weakness W1 and Negatives N3 and N1**
>
> ### Reviewer RNJJ - W1
> *W1) Does not discuss the choice between T2I-based video models vs open-source T2V models (such as WAN, LTX, etc.). I am aware of their limitations, but I do think it deserves discussion rather than a single line.*
>
> ### Answer RNJJ W1
> We thank the reviewer for highlighting this point.
>
> The choice between T2I-based and T2V-based approaches is primarily governed by two practical considerations: supported video length and computational cost.
> A distinctive aspect of our approach is its applicability to long, dynamic video sequences. To our knowledge, we are among the first to evaluate our video editing approach on sequences exceeding 300 frames with substantial camera motion, such as driving scenes from BDD100K.
> Beyond length constraints, memory limitations remain a fundamental bottleneck when loading long video sequences, which is precisely what motivates our zero-shot adaptive sampling methodology. Our approach is designed to be scalable at low computational cost, while remaining applicable across a wide range of video datasets and scenarios.
>
>
> At the time of writing, we only found available access (shared weights) for three open-source video generation/editing models: AnimateDiff published at ICLR 2024, Wan2.1 and LTX Video only on arXiv.
> AnimateDiff is limited to 16-frames processing and was published in 2024.
> We performed a quantitative comparison with our method in the paragraph "Short-term editing" of Section 5.2 ("Results") of the main paper.
> LTX Video and Wan2.1 are also limited to a maximum input length of around 100-frames videos.
> A possibility to extend the length would be to edit videos by block, without the assurance of long-term consistency (which is the main problem we address in this paper).
> Lastly, available editing models such as LTX Video and Wan2.1-VACE are using $\sim$13-14B parameters which is not reasonably comparable to our lightweight approach.
>
> We acknowledge that T2V models would benefit from native temporal reasoning and will likely close the quality gap as hardware capabilities continue to improve.
> However, in the zero-shot long-video regime, T2I-based methods currently can still offer a more practical trade-off between editability, scalability and computational efficiency.
>
> The introduction paragraph on T2I backbone choice to address was modified to include elements from this discussion.
>
> ---
>
> ### Reviewer RNJJ - N3
> *N3) Using SD1.5 in 2025-2026 is very outdated and makes it even harder to understand the scaling of the properties discussed in the paper. Can it be applied to newer models while getting the same benefits? Unsure!*
>
> ### Answer RNJJ N3
> We chose Stable Diffusion 1.5 as our backbone primarily to ensure fair comparisons with competing methods (VidToMe, RAVE, AnimateDiff, VideoDirector and VideoGrain), all of which are built on this SD 1.5 backbone, even those published in 2025 (VideoDirector and VideoGrain).
>
> The core contributions of AMAC operate at the level of the diffusion inference pipeline and do not rely on any SD1.5 specific architectural properties.
> We would therefore expect the same benefits to carry over to more recent architectures where temporal coherence over long sequences remains an open challenge.
>
> ---
>
> ### Reviewer RNJJ - N1
> *N1) Ablation only shows quantitative numbers. Why not use a human study to prove this point as well, if quant metrics are not that reliable?*
>
> ### Answer RNJJ N1
>
> The ablation study on the sampling strategy and token merging presented in Section 5.3 of the main paper shows quantitative scores in Table 4.
> We acknowledge the reviewer's valid point that quantitative metrics alone may not fully capture perceptual quality differences.
>
> Conducting a rigorous user study is however non-trivial, as it requires a sufficient number of participants and carefully designed stimuli to yield statistically meaningful results.
> Given our constrained evaluator pool, we prioritized the human study for our main comparative results, where the perceptual impact of design choices is most prominent.
>
> Conducting a comprehensive user study is non-trivial, as it requires a significant number of participants to obtain meaningful results.
> Given our limited pool of evaluators, we prioritized the human study on our main results.
>
> Additionally, the quantitative metrics we employ are those adopted by prior works in the field, ensuring comparability with the existing literature.
> While we agree that human evaluation of ablation conditions would be a valuable complement, we note that the ablation configurations being compared are numerous and more subtle, making perceptual studies both longer and harder for participants to evaluate.

---

> ### Author Response · Authors · 2026-03-03
> **Answers to reviewer RNJJ: Negatives N2 and N4, Weaknesses W2 and W3, and Broader Impact Statement**
>
> ### Reviewer RNJJ - N2 and W2
>
> *N2) Comparison to very recent methods, such as those mentioned in this line: "Furthermore, most T2V editing methods are currently not open source, for example, Gao et al. (2025); Yang et al. (2025); Wang et al. (2025a); Zhu et al. (2025); Zhang et al. (2025)", is very important for understanding the impact of this contribution. This makes it very difficult to make a strong decision.*
>
> *W2) Many of the cited works are not available and thus not easily comparable. This makes it harder to read, since the discussion runs alongside the cited works, but the results don't compare to them.*
>
> ### Answer RNJJ N2 and W2
> We are fully in agreement in the reviewer's concern regarding the comparison with closed-source methods.
> We would like to compare with all the cited methods, but unfortunately, the lack of publicly available code for these models makes direct quantitative comparison infeasible.
>
> ---
>
> ### Reviewer RNJJ - N4
> *N4) Explanation of the choice of similarity proxy (similarity between VAE encoded frames).*
>
> ### Answer RNJJ N4
> We chose a content-based similarity proxy rather than a temporal proximity measure, because our goal is to group frames with visually similar content regardless of their temporal position in the video.
> For instance, in scenarios where the camera moves away and then returns to a similar viewpoint, temporally distant frames may share highly similar visual content and thus benefit from being processed together.
> Using VAE latent similarity allows us to capture this structural redundancy in a computationally efficient manner.
>
> ---
>
> ### Reviewer RNJJ - W3
> *W3) Weak Proxy for frame similarity: Based on my understanding, the frame similarity is the key guiding feature in AMAC. Wouldn't it be better if a more semantic feature were used for this choice, such as CLIP similarity?*
>
> ### Answer RNJJ W3
> We want to thank the reviewer for this interesting question. We chose VAE latent similarity over CLIP embeddings for two main reasons.
>
> First, CLIP is trained on text-image pairs and is designed to capture high-level semantic alignment between modalities rather than fine-grained visual differences between video frames.
> Within a single video, consecutive or nearby frames tend to be semantically very similar, for instance two frames both depicting a car on a road would receive near-identical CLIP embeddings despite potentially differing in viewpoint, lighting or spatial configuration.
> VAE latent representations, by contrast, are sensitive to these low-level structural differences, making them better suited for capturing the frame-to-frame visual variation that guides our permutation strategy.
>
> Second, since our backbone already encodes frames into the VAE latent space as part of the standard diffusion pipeline, we can leverage these representations directly at no additional cost.
> Extracting CLIP embeddings would require a separate encoding step, introducing computational overhead at the pre-processing stage without a proportional gain in permutation quality.
> More broadly, the core principle of AMAC is to derive an adaptive permutation that reflects the relative similarity structure of frames, making the precise choice of similarity metric a secondary concern.
> VAE latent similarity satisfies this requirement efficiently and in a manner well-aligned with the permutation objective.
>
> ---
>
> ### Reviewer RNJJ - Broader Impact Statement
> *BIS) The paper does not explicitly include a Broader Impact Statement. While it is a technical contribution to video editing, text-to-video models carry inherent risks regarding deepfakes and the generation of misleading content.
> Recommendation: The authors should add a brief statement acknowledging that while AMAC improves creative editing, it could potentially be used to create more convincing synthetic videos, necessitating the continued development of watermarking or detection tools.*
>
> ### Answer RNJJ BIS
> While our method is primarily designed for industrial and artistic video editing, we recognize that, like any video generation or editing framework, it could potentially be misused to produce misleading or deceptive content, such as deepfakes. We strongly advocate for the parallel development of robust detection and watermarking tools to mitigate such risks, and we encourage the community to use such methods responsibly.
>
> We will add a Broader Impact Statement at the end of the Conclusion of the main paper acknowledging the potential risks associated with video editing methods.

---

### Decision · Action_Editor_EaAP · 2026-04-09

**Recommendation:** Accept as is

**Audience:**

Yes

**Audience Explanation:**

The TMLR audience interested in generative modeling, diffusion processes, and video synthesis will find the paper's formulation of frame sampling as a stochastic permutation process highly relevant. The adaptive sampling approach offers a valuable, training-free methodology for handling long video sequences, which remains an active and challenging area of research.

**Claims And Evidence:**

Yes

**Claims Explanation:**

Initially, reviewers raised valid concerns regarding the scale of the user study, the lack of computational overhead analysis, and specific physical/semantic failure cases (such as directional logic errors and blurriness). During the rebuttal phase, the authors successfully addressed these issues by expanding the user study to include the DAVIS dataset with 46 new participants, adding a detailed runtime/memory comparison against baselines, and transparently documenting the failure cases inherent to the stochastic permutation strategy and the SD1.5 backbone. With these revisions, the claims of improving temporal coherence in zero-shot text-to-video editing are now well-supported by both quantitative metrics and expanded qualitative human evaluation.